# Sm: enhanced localization in Multiple Instance Learning for medical imaging classification

**Francisco M. Castro-Macías**
CITIC-UGR
Dept. of Comp. Science and A. I.
University of Granada

**Pablo Morales-Álvarez**
Dept. of Statistics and Operations Research
CITIC-UGR
University of Granada

**Yunan Wu**
Dept. of Elect. and Comp. Engineering
Northwestern University

**Rafael Molina**
Dept. of Comp. Science and A. I.
University of Granada

**Aggelos K. Katsaggelos**
Dept. of Elect. and Comp. Engineering
Northwestern University

## Abstract

Multiple Instance Learning (MIL) is widely used in medical imaging classification to reduce the labeling effort. While only bag labels are available for training, one typically seeks predictions at both bag and instance levels (classification and localization tasks, respectively). Early MIL methods treated the instances in a bag independently. Recent methods account for global and local dependencies among instances. Although they have yielded excellent results in classification, their performance in terms of localization is comparatively limited. We argue that these models have been designed to target the classification task, while implications at the instance level have not been deeply investigated. Motivated by a simple observation – that neighboring instances are likely to have the same label – we propose a novel, principled, and flexible mechanism to model local dependencies. It can be used alone or combined with any mechanism to model global dependencies (e.g., transformers). A thorough empirical validation shows that our module leads to state-of-the-art performance in localization while being competitive or superior in classification. Our code is at `https://github.com/Franblueee/SmMIL`.

## 1 Introduction

Over the last decades, medical imaging classification has benefited from advances in deep learning [35, 44]. However, the performance of these methods drops when the number of labeled samples is low, which is common in real-world medical scenarios [1]. To overcome this, Multiple Instance Learning (MIL) has emerged as a popular weakly supervised approach [14, 8, 12].

In MIL, instances are arranged in bags. At train time, a label is available for the entire bag, while the instance labels remain unknown. The goal is to train a method that, given a test bag, can predict both at bag and instance levels (classification and localization tasks, respectively). This paradigm is well suited to the medical imaging domain [28]. In cancer detection from Whole Slide Images (WSIs), the WSI represents the bag, and the patches are the instances. In intracranial hemorrhage detection from Computerized Tomographic (CT) scans, the full scan represents the bag, and the slices at different heights are the instances. In these scenarios, making accurate predictions of instance

labels is extremely important from a clinical viewpoint, as it translates into pinpointing the location of the lesion [7].

Most successful approaches in MIL build on the attention-based pooling [17], a permutation-invariant operator that assigns an attention value to each instance independently. This method has been extended in different ways while maintaining the permutation-invariant property [21, 25, 39]. The aforementioned works pose a problem: the dependencies between the instances, which are important when making a diagnosis, are ignored. To account for this, TransMIL [32] proposed to model global dependencies using a transformer encoder. The idea is to use the self-attention mechanism to introduce interactions between each pair of instances. Based on it, other transformer-based approaches have emerged, also focusing on global dependencies [9, 22, 37]. More recently, several works have also incorporated natural local interactions, which are those between neighboring instances [14, 40, 41].

Although these methods accounting for dependencies have resulted in excellent performance at the bag level, the evaluation at the instance level has received less attention and the results are not comparatively good so far, see the very recent [14]. In this work, we argue that recent MIL methods have been designed with the classification task in mind, and we propose a new model that focuses on both the classification and localization tasks. Specifically, we propose a novel and theoretically grounded mechanism to introduce local dependencies, hereafter referred to as *the smooth operator* Sm. This is a flexible module that can be used alone on top of classical MIL approaches, or in combination with transformers to also account for global dependencies. In both cases, we show that the proposed operator achieves state-of-the-art localization results while being competitive in classification. We compare against a total amount of eight methods, including very recent ones [14, 40]. We utilize three different datasets of different nature and size, covering two different medical imaging problems (cancer detection in WSI images and hemorrhage detection in CT scans).

Our main contributions are: (i) we provide a unified view of current deep MIL approaches; (ii) we propose a principled mechanism to introduce local interactions, which is a modular component that can be combined or not with global interactions; and (iii) we evaluate up to eight state-of-the-art MIL methods on three real-world MIL datasets in both classification and localization tasks, showing that the proposed method stands out in localization while being competitive or superior in classification.

## 2   Related work

In this work, we tackle the localization task in deep MIL methods using existing concepts and techniques from deep MIL and Graph Neural Networks (GNNs) theory.

**Deep Multiple Instance Learning.** As explained by Song et al. [34], deep MIL methods can be divided into two broad categories, namely instance-based or embedding-based, depending on the level at which a specific *aggregation* operator is applied. In this paper, we focus on the embedding-based category, and in particular attention-based ones.

Ilse et al. [17] proposed attention-based pooling to weigh each instance in the bag. To improve it, different modifications were proposed, including the use of clustering layers [25], grouping the instances in pseudo-bags [39], and using similarity measures to critical instances to compute the attention values [21]. However, these methods ignore the existing dependencies between the instances in a bag. To address this, Shao et al. [32] proposed to use a transformer-based architecture and the PPEG position encoding module. This has been extended with different transformer variations, including the deformable transformer architecture [22], hierarchical attention [37], and regional sampling [9]. Recently, these methods have been improved to include spatial information in different ways, including the use of a Graph Convolutional Network (GCN) before the transformer [41], a neighbor-constrained attention module [14], and a spatial-encoding transformer architecture [40].

In the studies mentioned above, the objective is to obtain increasingly better bag-level results, while the evaluation at the instance level is usually performed qualitatively. In contrast, our work addresses both the instance localization task and the bag classification task, as both are of great importance for making a diagnosis. Moreover, our work is not limited to WSI classification; it is also valid for other medical imaging modalities.

**Graph Neural Networks.** Our motivation — that neighboring instances are likely to have the same label — is a well-established assumption within the machine learning community, often referred to as the cluster assumption [10, 31]. Since leveraged in 1984 by Ripley [30] in the context of spatial

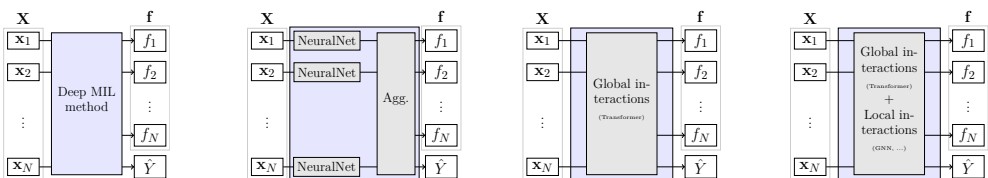

(a) General deep MIL model.

(b) Instances are treated independently.

(c) Only global interactions.

(d) Global and local interactions.

Figure 1: (a) Unified view of deep MIL models. Depending on how instances interact with each other in (a), we devise three different families of methods: (b), (c), (d).

statistics, it has been extensively used in spectral clustering [27], semi-supervised learning on graphs [3], and recently in GNNs [20]. Our work builds upon seminal works in these areas.

The proposed smooth operator is derived considering a Dirichlet energy minimization problem, similar to the work by Zhou and Schölkopf [42] and Zhou et al. [43]. This approach has been employed in recent years to obtain new GNN models, including the $p$-Laplacian layer [15], and PPNP layer [16]. Moreover, the Dirichlet energy has been studied in the context of GNNs to analyze the over-smoothing phenomenon [6, 23]. In this regard, our bound on the decrease of the Dirichlet energy is analogous to the result derived by Li et al. [23] to study over-smoothing for GCNs. Our result, however, holds for the proposed mechanism, of which the graph convolutional layer is a special type.

## 3    Background: A unified view of deep MIL approaches

We first describe the binary MIL problem tackled in this paper. Then, we provide a unified view of the most popular deep MIL methods. As explained in Sec. 2, we focus on embedding-based approaches.

In MIL, the training set consists of pairs of the form $(\mathbf{X}, Y)$, where $\mathbf{X} \in \mathbb{R}^{N \times P}$ is a bag of instances and $Y \in \{0, 1\}$ is the bag label. We write $\mathbf{X} = [\mathbf{x}_1, \ldots, \mathbf{x}_N]^T \in \mathbb{R}^{N \times P}$, where $\mathbf{x}_n \in \mathbb{R}^P$ are the instances. Each instance $\mathbf{x}_n$ is associated to a label $y_n \in \{0, 1\}$, *not available during training*. It is assumed that $Y = \max \{y_1, \ldots, y_N\}$, i.e., a bag $\mathbf{X}$ is considered positive if and only if there is at least one positive instance in the bag.

Given a previously unseen bag (e.g., a medical image), the goal at test time is to: i) predict the bag label (classification task) and ii) obtain predictions or estimates for the instance labels (localization task). In general, deep MIL models output a bag-level prediction $\hat{Y}$, as well as instance-level scalars $f_n$ that are used for instance-level prediction. This general process is depicted in Fig. 1a. In many approaches, these $f_n$ are the so-called *attention values* (e.g., ABMIL [17], TransMIL [32], CAMIL [14]), but they can be obtained in different ways (e.g., through GraphCAM in GTP [41]). Within the general process in Fig. 1a, deep MIL models can be categorized into three families, depending on how instances interact with each other, see Fig. 1b, Fig. 1c, and Fig. 1d.

In the first family, shown in Fig. 1b, the instances are encoded *independently* and then aggregated. The well-known ABMIL [17] fits in this paradigm. Subsequent works introduce slight modifications to ABMIL, while still encoding each instance *independently* [21, 25, 39]. ABMIL, on which we will rely to introduce our model, is depicted in Fig. 3a. First, a bag of embeddings $\mathbf{H} = [\mathbf{h}_1, \ldots, \mathbf{h}_N] \in \mathbb{R}^{N \times D}$ is obtained by applying a neural network independently to each instance. Then, the attention-based pooling computes the attention values $\mathbf{f}$ and the bag embedding $\mathbf{z}$ according to

$$\mathbf{F} = \tanh\left(\mathbf{H}\mathbf{W}^\top\right), \quad \mathbf{f} = \mathbf{F}\mathbf{w}, \tag{1}$$

$$\mathbf{z} = \mathrm{AttPool}\left(\mathbf{H}\right) = \mathbf{H}^\top \mathrm{Softmax}\left(\mathbf{f}\right), \tag{2}$$

where $\mathbf{W} \in \mathbb{R}^{L \times D}$, $\mathbf{w} \in \mathbb{R}^L$ are trainable weights. Last, $\hat{Y}$ is obtained by applying a linear classifier on $\mathbf{z}$.

The second family accounts for *global* interactions between instances, possibly long-range ones, see Fig. 1c. These works treat instances as tokens that interact through the self-attention mechanism. This way, global interactions between instances are learned. One of the most popular approaches in this family is TransMIL [32], which was later extended in different directions [9, 22]. The third family complements the previous one with *local* interactions defined by a fixed neighborhood, see

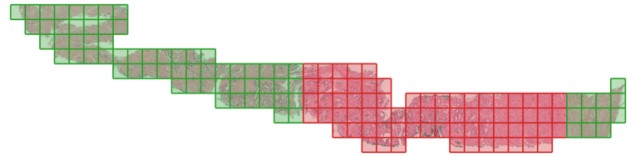

(a) Labelled patches in a WSI.

(b) Labelled slices in a CT scan.

Figure 2: WSIs are divided into patches. CT scans are provided as slices. They often show spatial dependencies: in a WSI, a patch is usually surrounded by patches with the same label, while in a CT scan, a slice is usually surrounded by slices with the same label. The red color indicates malignant/hemorrhage patches/slices.

Fig. 1d. They differ in how local interactions are represented, e.g., as a graph in CAMIL [14] and GTP [41], or using a position-encoded feature map in SETMIL [40].

In most of these works, the localization task is assessed qualitatively, e.g., by visually comparing the attention maps. This contrasts with the classification task, which is always evaluated quantitatively. As evidenced by Fourkioti et al. [14], this has translated into comparatively poor performance in terms of localization. We notice that current models have been designed to target the classification task, and they excel at that. However, their model design is not as careful about the instance-level implications. For example, CAMIL [14] does not leverage any local information to obtain the instance-level attention values. Indeed, from their Eq. (8) one deduces that the $a_i$ values are obtained from the tile representations $\mathbf{t}_i$, which have not undergone any local interaction. Observe that local interactions take place in Eq. (4) and Eq. (5) in their paper, but these only affect the bag-level predictions, not the instance-level ones. Similarly, GTP [41] introduces local interactions through an off-the-shelf graph convolutional layer, the effect of which is not investigated at the instance level. In the following section, we propose a principled approach to account for meaningful local interactions based on the Dirichlet energy. The idea is motivated by a natural property often observed in the instance-level labels of medical images: the similarity between neighboring instances.

## 4   Method: Introducing smoothness in the attention values

In medical imaging, instance labels are *a priori* expected to exhibit local dependencies with their neighboring instances: an instance is likely to be surrounded by instances with the same label, see Fig. 2. Recall that attention values are commonly used as a proxy to estimate these labels, so they should inherit this property. Based on these observations, our intuitive idea is to favor a *smoothness* property on the attention values. To this end, Sec. 4.1 formalizes the notion of smoothness through the Dirichlet energy. Sec. 4.2 presents the proposed smoothing operator Sm, which encourages smoothness as well as fidelity to the original signal. Sec. 4.3 proposes how to leverage Sm in the context of MIL, both in combination with global interactions (via transformers), and without them. We will build on top of the well-known and simple ABMIL to isolate the effect of Sm and avoid over-sophisticated models.

### 4.1   Modelling the smoothness

We represent each bag as a graph, where the nodes are the instances and the edges represent the spatial connectivity between instances. Formally, we suppose that each bag $\mathbf{X} \in \mathbb{R}^{N \times D}$ has been assigned an adjacency matrix $\mathbf{A} = [A_{ij}] \in \mathbb{R}^{N \times N}$, defined by $A_{ij} > 0$ if instances $\mathbf{x}_i$ and $\mathbf{x}_j$ are neighbors, and $A_{ij} = 0$ otherwise. We assume that the adjacency matrix is symmetric, i.e. $A_{ij} = A_{ji}$.

The *Dirichlet energy* is a well-known functional that measures the variability of a function defined on a graph [42, 43]. In our case, we think of this function as the attention values $\mathbf{f} \in \mathbb{R}^N$, recall Fig. 1a. As we shall see below, it will be necessary to define the Dirichlet energy for multivariate graph functions. Given a multivariate graph function $\mathbf{U} = [\mathbf{u}_1, \ldots, \mathbf{u}_N]^\top \in \mathbb{R}^{N \times D}$ defined on the bag graph, the Dirichlet energy of $\mathbf{U}$ is given by

$$\mathcal{E}_D\left(\mathbf{U}\right) = \tfrac{1}{2} \sum_{i=1}^N \sum_{j=1}^N A_{ij} \left\| \mathbf{u}_i - \mathbf{u}_j \right\|_2^2 = \mathrm{Trace}\left(\mathbf{U}^\top \mathbf{L} \mathbf{U}\right), \tag{3}$$

where $\|\cdot\|_2$ denotes the Euclidean norm, $\mathbf{L}$ is the graph Laplacian matrix $\mathbf{L} = \mathbf{D} - \mathbf{A}$, $\mathbf{D} \in \mathbb{R}^{N \times N}$ is the degree matrix, $\mathbf{D} = \mathrm{Diag}\,(D_1, \ldots, D_N)$, $D_n = \sum_i A_{ni}$. When $D = 1$ we obtain the definition for univariate graph functions, such as the attention values $\mathbf{f}$.

**Bounding $\mathcal{E}_D$ on the attention values.** In most deep MIL approaches, the attention values $\mathbf{f}$ are obtained applying a neural network to instance-level features. One example is ABMIL [17], which uses a two-layer perceptron defined by Eq. 1. Noting that $\mathrm{tanh}$ is a Lipschitz function with Lipschitz constant equal to 1, we arrive at the following chain of inequalities

$$\mathcal{E}_D\,(\mathbf{f}) \leq \|\mathbf{w}\|_2^2\,\mathcal{E}_D\,(\mathbf{F}) \leq \|\mathbf{w}\|_2^2\,\|\mathbf{W}\|_2^2\,\mathcal{E}_D\,(\mathbf{H})\,, \tag{4}$$

where $\|\cdot\|_2$ denotes the spectral norm. A more general result holds in the general case of an arbitrary multi-layer perceptron, see Appendix A for a proof. The above chain of inequalities tells us that if we want $\mathcal{E}_D\,(\mathbf{f})$ to be low, *we can act on $\mathbf{f}$ itself or on previous layers (e.g., on $\mathbf{F}$ or on $\mathbf{H}$)*, constraining the norm of the trainable weights to remain constant. This constraint can be achieved using spectral normalization [26], and we study its influence in Sec. B.3. In the next subsection, we propose an operator that can be used on any of these levels ($\mathbf{f}$, $\mathbf{F}$, $\mathbf{H}$) to reduce the Dirichlet energy of its output.

## 4.2   The smooth operator

Our goal now turns into finding an operator $\mathtt{Sm} : \mathbb{R}^{N \times D} \to \mathbb{R}^{N \times D}$ that, given a bag graph multivariate function $\mathbf{U} \in \mathbb{R}^{N \times D}$, returns another bag graph multivariate function $\mathtt{Sm}\,(\mathbf{U}) \in \mathbb{R}^{N \times D}$ such that its Dirichlet energy is lower without losing the information present in the original $\mathbf{U}$. Following seminal works [42, 43], we cast this as an optimization problem,

$$\mathtt{Sm}\,(\mathbf{U}) = \arg\min_{\mathbf{G}} \mathcal{E}\,(\mathbf{G})\,, \tag{5}$$

$$\mathcal{E}\,(\mathbf{G}) = \alpha \mathcal{E}_D\,(\mathbf{G}) + (1 - \alpha)\,\|\mathbf{U} - \mathbf{G}\|_F^2\,, \tag{6}$$

where $\alpha \in [0, 1)$ accounts for the trade off between both terms, and $\|\cdot\|_F$ denotes the Frobenius norm. The first term in the above equation penalizes functions with too much variability, while the second term penalizes functions that differ too much from the original $\mathbf{U}$. Note that this can be interpreted as a maximum a posteriori formulation, where the first term corresponds to the prior distribution and the second to the observation model, see [30]. The objective function $\mathcal{E}$ is strictly convex, and therefore admits a unique solution, given by

$$\mathtt{Sm}\,(\mathbf{U}) = (\mathbf{I} + \gamma\mathbf{L})^{-1}\,\mathbf{U}, \tag{7}$$

where $\gamma = \alpha/(1 - \alpha)$. Unfortunately, the expression in Eq. (7), although elegant, incurs prohibitive computational and memory costs, especially when the number of instances in the bag is large (which is the case of WSIs). Instead, we can take an iterative approach, defining $\mathtt{Sm}\,(\mathbf{U}) = \mathbf{G}(T)$, with

$$\mathbf{G}(0) = \mathbf{U}; \quad \mathbf{G}(t) = \alpha\,(\mathbf{I} - \mathbf{L})\,\mathbf{G}(t - 1) + (1 - \alpha)\,\mathbf{U}, \quad t \in \{1, \ldots, T\}\,. \tag{8}$$

As demonstrated by Zhou et al. [43], the sequence $\{\mathbf{G}(t)\}$ converges to the optimal solution in Eq. 7. As studied by Gasteiger et al. [16], it is enough to use a small number of iterations $T$ to approximate the exact solution. Therefore, in this work, we will adopt the iterative approach described by Eq. 8. Based on previous work [16], we will use $T = 10$, and $\alpha$ will be set as a trainable parameter initialized at $\alpha = 0.5$. See Sec. B.3 for a study on the effects of these hyperparameters and Fig. 12 for a visual comparison of the effect that $\alpha$ has on the attention maps.

**Theoretical guarantees via the normalized Laplacian.** We present a result that informs us about the rate at which the Dirichlet energy decreases when applying $\mathtt{Sm}$. Let us define $\lambda_\gamma^* = \max\left\{(1 + \gamma\lambda_n)^{-2} : \lambda_n \in \Lambda \setminus \{0\}\right\}$, where $\Lambda = \{\lambda_1, \ldots, \lambda_N\}$ are the eigenvalues of the bag graph Laplacian matrix. Then, we have the following inequality,

$$\mathcal{E}_D\,(\mathtt{Sm}\,(\mathbf{U})) \leq \lambda_\gamma^* \mathcal{E}_D\,(\mathbf{U})\,. \tag{9}$$

The proof is inspired by Cai and Wang [6], see Appendix A. If $\lambda_\gamma^* < 1$, then the smooth operator effectively decreases the Dirichlet energy. If we replace the Laplacian matrix by the normalized Laplacian matrix, $\tilde{\mathbf{L}} = \mathbf{D}^{-1/2}\mathbf{L}\mathbf{D}^{-1/2}$, it is known that its eigenvalues lie in the interval $[0, 2)$, and then $\lambda_\gamma^* < 1$ holds. This motivates the use of the normalized Laplacian in our experiments.

The smooth operator $\mathtt{Sm}$ only introduces one parameter to be estimated, $\alpha$. Also, it is differentiable with respect to its input. Therefore, it can be integrated into simple attention-based MIL models, such as ABMIL, to account for local dependencies.

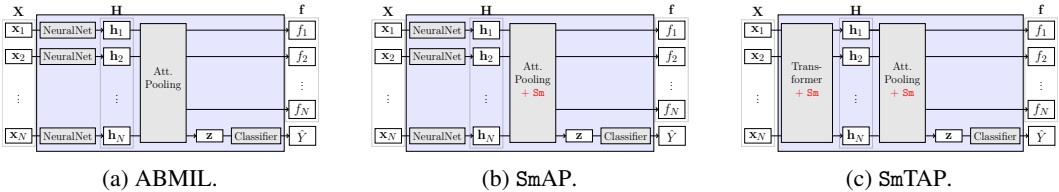

|  (a) ABMIL. | (b) SmAP. | (c) SmTAP. |

Figure 3: Smooth Attention Multiple Instance Learning. (a) The well-known model in [17], which we build upon. (b): only local interactions are considered by applying the proposed smooth operator Sm in the aggregation part. (c): both global and local interactions are considered by applying Sm both in the transformer and in the aggregation parts.

## 4.3 The proposed model

Here we propose how to leverage the operator Sm in the context of MIL. We build on top of the well-known ABMIL. First, we introduce SmAP, which integrates ABMIL with Sm and only accounts for local interactions. Second, we introduce SmTAP, which equips SmAP with a transformer encoder to account for global dependencies. The proposed models are depicted in Fig. 3b and Fig. 3c. The details about the architecture we have used can be found in Sec. B.2.

**SmAP: Smooth Attention Pooling.** This is represented in Fig. 3b. First, the bag of embeddings $\mathbf{H}$ is obtained as in ABMIL [17], i.e. treating the instances independently. Then, the operator Sm is integrated within the attention pooling. Based on Eq. 4, this can be done on the attention values themselves or on previous representations. We consider three different variants: SmAP-late, SmAP-mid, SmAP-early. They act, respectively, on $\mathbf{f}$ (the attention values themselves), on $\mathbf{F}$ (i.e. before entering the last layer), and on $\mathbf{H}$ (i.e. before entering the attention-based pooling). Formally,

$$\text{late:} \quad \mathbf{f} = \text{Sm}\left(\tanh\left(\mathbf{HW}^{\top}\right)\mathbf{w}\right), \tag{10}$$

$$\text{mid:} \quad \mathbf{f} = \tanh\left(\text{Sm}\left(\mathbf{HW}^{\top}\right)\right)\mathbf{w}, \tag{11}$$

$$\text{early:} \quad \mathbf{z} = \text{AttPool}\left(\text{Sm}\left(\mathbf{H}\right)\right), \tag{12}$$

While SmAP-late and SmAP-mid act on the computation of the attention values, SmAP-early acts on the embedding that is passed to the attention-based pooling, see Fig. 8 in Appendix C. We use SmAP-early by default. Sec. 5.3 shows that results do not differ much among configurations.

**SmTAP: Smooth Transformer Attention Pooling.** This is represented in Fig. 3c. The only difference with SmAP is that the neural network acting independently on the instance embeddings is replaced by a transformer encoder to account for global dependencies. Based on the idea that smoothness can be imposed at previous locations, recall Eq. 4, we propose to also apply Sm to the transformer output:

$$\mathbf{H} = \text{Sm}\left(\text{Softmax}\left(q\left(\mathbf{X}\right)k\left(\mathbf{X}\right)^{\top}\right)v\left(\mathbf{X}\right)\right), \tag{13}$$

where $q$, $k$, and $v$ are the standard queries, keys, and values in the dot product self-attention [4]. Notice that SmTAP uses Sm in two places: the first after the transformer encoder and the second in the aggregator. Naturally, one could think of other variants that use Sm in only one place or the other. In Sec. 5.3 we ablate these different configurations, leading to similar results.

## 5 Experiments

We validate the proposed Sm in three medical MIL datasets: RSNA [13], PANDA [5], and CAME-LYON16 [2]. We evaluate the effectiveness of our approach by a quantitative and qualitative analysis. All experiments have been conducted under fair and reproducible conditions. Details on the datasets and experimental setup can be found in Appendix B. The code is available at https://github.com/Franblueee/SmMIL.

We compare our approaches with state-of-the-art deep MIL methods. We consider two groups of methods, depending on the presence/absence of a transformer block to model global dependencies. In the first group, we include those models that do not use this block: the proposed SmAP, ABMIL [17], CLAM [25], DSMIL [21], and DFTD-MIL [39]. The second group consists of models that do use the transformer encoder: the proposed SmTAP, TransMIL [32], SETMIL [40], GTP [41], and CAMIL [14]. These groups ensure a fair comparison in terms of model capabilities and complexity.

Table 1: Localization results (mean and standard deviation from five independent runs). The best is in bold and the second-best is underlined. (↓)/(↑) means lower/higher is better. The proposed operator improves the localization results in all three datasets and both with and without global interactions. It ranks first in eight out of twelve dataset-score pairs.

| | | RSNA | | PANDA | | CAMELYON16 | | |
|---|---|---|---|---|---|---|---|---|
| | | AUROC (↑) | F1 (↑) | AUROC (↑) | F1 (↑) | AUROC (↑) | F1 (↑) | Rank (↓) |
| Without global interactions | SmAP | $\underline{0.798}_{0.033}$ | $\underline{0.477}_{0.014}$ | $\mathbf{0.799}_{0.005}$ | $\underline{0.635}_{0.006}$ | $\mathbf{0.960}_{0.007}$ | $\mathbf{0.840}_{0.053}$ | $\mathbf{1.500}_{0.548}$ |
| | ABMIL | $\mathbf{0.806}_{0.012}$ | $\mathbf{0.486}_{0.033}$ | $0.768_{0.002}$ | $0.602_{0.004}$ | $0.819_{0.074}$ | $0.766_{0.060}$ | $\underline{2.500}_{1.225}$ |
| | CLAM | $0.523_{0.069}$ | $0.076_{0.154}$ | $0.727_{0.046}$ | $0.568_{0.038}$ | $0.849_{0.044}$ | $\underline{0.821}_{0.046}$ | $4.167_{1.329}$ |
| | DSMIL | $0.554_{0.004}$ | $0.180_{0.000}$ | $0.765_{0.008}$ | $0.598_{0.006}$ | $0.760_{0.070}$ | $0.654_{0.183}$ | $4.333_{0.516}$ |
| | DFTD-MIL | $0.747_{0.070}$ | $0.453_{0.194}$ | $\underline{0.795}_{0.004}$ | $\mathbf{0.637}_{0.006}$ | $\underline{0.884}_{0.002}$ | $0.742_{0.040}$ | $2.500_{1.049}$ |
| With global interactions | SmTAP | $\mathbf{0.767}_{0.046}$ | $\mathbf{0.474}_{0.023}$ | $\mathbf{0.790}_{0.007}$ | $0.622_{0.01}$ | $\mathbf{0.789}_{0.008}$ | $\mathbf{0.600}_{0.067}$ | $\mathbf{1.500}_{1.225}$ |
| | TransMIL | $0.732_{0.013}$ | $\underline{0.471}_{0.014}$ | $0.751_{0.011}$ | $\mathbf{0.636}_{0.008}$ | $\underline{0.781}_{0.024}$ | $0.127_{0.078}$ | $3.083_{1.429}$ |
| | SETMIL | $0.726_{0.025}$ | $0.438_{0.027}$ | $0.774_{0.007}$ | $0.631_{0.010}$ | $0.615_{0.231}$ | $0.134_{0.267}$ | $3.667_{0.816}$ |
| | GTP | $0.736_{0.017}$ | $0.425_{0.018}$ | $0.768_{0.022}$ | $\underline{0.636}_{0.011}$ | $0.442_{0.091}$ | $0.037_{0.036}$ | $3.917_{1.429}$ |
| | CAMIL | $\underline{0.760}_{0.036}$ | $0.456_{0.013}$ | $\underline{0.785}_{0.011}$ | $0.621_{0.013}$ | $0.742_{0.028}$ | $\underline{0.479}_{0.175}$ | $\underline{2.833}_{1.169}$ |

In Sec. B.3 we report the results of three more methods: DeepGraphSurv [24], PathGCN [11], and IIBMIL [29]. Note that the performance obtained by these methods does not affect the conclusions we will obtain in this section.

In Sec. 5.1 we consider the localization task. In Sec. 5.2 we turn to the classification task. Sec. 5.3 shows an ablation study on how different uses of the smooth operator affect the proposed model.

## 5.1 Localization: instance level results

In this subsection, we analyze the ability of each model to predict the label of the instances inside a bag. As explained in Sec. 3, deep MIL models assign a scalar value $f_n$ to each instance $\mathbf{x}_n$, see Fig. 1a. Although these can be obtained in different ways, for simplicity we will refer to them as *attention values*. Thus, we compare the attention values with the ground truth instance labels, which are available for the test set only for evaluation purposes.

**Quantitative analysis.** We analyze the performance of each method using the area under the ROC curve (AUROC) and the F1 score. Note that a critical hyperparameter for the latter is the threshold used on $f_n$ to determine the label of each instance. To ensure a fair comparison, we compute the optimal threshold for each method using the validation set. As a general summary, we also report the average rank achieved by each model across metrics and datasets.

The results are shown in Table 1. We find that using Sm provides the best performance overall, placing as the best or second-best within each group. Only in RSNA the proposed SmAP is outperformed by ABMIL. We attribute this to the fact that the bag graphs in CT scans are not as complex as in WSIs, and therefore the local interactions are not as meaningful. Note that the performance gain is particularly significant on CAMELYON16, where the bags have a larger number of instances, the graphs are much denser and the imbalance between positive and negative instances is more severe. Notably, SmTAP significantly outperforms SETMIL, GTP, and CAMIL, which also model local dependencies. Contrary to our method, their design is focused on bag-level performance and it does not translate into meaningful instance-level properties.

**Attention histograms.** We examine the attention histograms produced by each model on the CAMELYON16 dataset. The corresponding figures for RSNA and PANDA can be found in Appendix C. In Fig. 4, we represent the frequency with which attention values are assigned to positive and negative instances, separately. An ideal instance classifier would place all the positive instances on the right and all the negative instances on the left. This illustrates why SmTAP and SmAP achieve such a good performance: they concentrate the negative instances to the left of the histogram while succeeding in grouping a large part of the positive instances to the right. TransMIL and GTP assign low attention values to both positive and negative instances. CAMIL is able to identify positive instances, but negative instances are assigned from intermediate to high attention values. CLAM and DSMIL assign low attention values to negative instances, but the distribution of the positive instances resembles a uniform and a normal distribution, respectively.

**Attention maps.** To visualize the localization differences, we show the attention maps generated by four of the transformer-based methods in a WSI from CAMELYON16, see Fig. 5. SmTAP attention

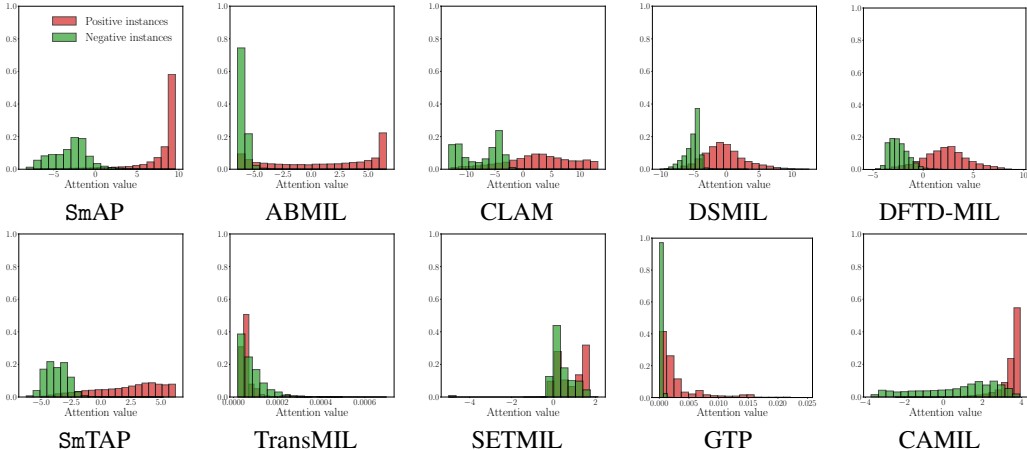

Figure 4: Attention histograms on CAMELYON16. First/second rows show models without/with global interactions. SmAP and SmTAP stand out at separating positive and negative instances.

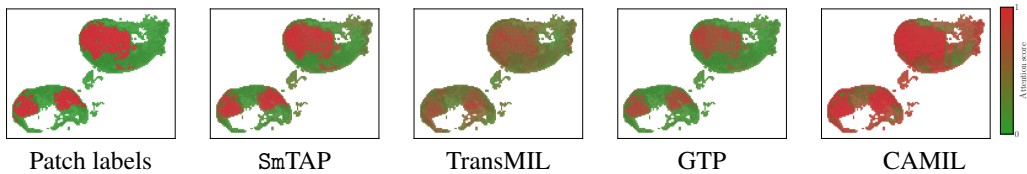

Figure 5: Attention maps on CAMELYON16. The novel SmTAP produces the most accurate one.

map resembles the most to the ground truth. As noted in Fig. 4, CAMIL assigns high attention values to both positive and negative instances. TransMIL and GTP pinpoint the regions of interest, but the attention is relatively low in those areas, which produces unclear boundaries, especially in the case of TransMIL. The attention maps for the rest of the methods and datasets are in Appendix C.

The results shown in this subsection validate the utility of the smooth operator at the instance level. They suggest that having smooth attention maps is a powerful inductive bias that improves the instance-level performance. In the following, we analyze its impact at the bag level.

## 5.2 Classification: bag level results.

In this subsection, we show that the use of the smooth operator does not deteriorate the bag classification results. On the contrary, in some cases, it improves them. Again, we focus on the AUROC and F1 scores, measured by comparing the true bag labels with the methods' bag label predictions. The threshold for the F1 score is $0.5$. We also report the mean rank achieved by each model.

Table 2 shows the results. The proposed models achieve the best performance overall. As in the localization task, ABMIL performs better than SmAP in RSNA. Again, we believe it to be a consequence of the CT scan's low-complexity structure. DFTD-MIL obtains the best result in CAMELYON16, but ranks second or third in the other two datasets. GTP and SETMIL outperform the proposed SmTAP in PANDA, but their performance significantly decreases in CAMELYON16, obtaining the worst results. Overall, our methods provide the most consistent performance, achieving an aggregated mean rank of 1.833.

## 5.3 Ablation study

The proposed Sm comes with different design choices and hyperparameters: the placement of Sm, the trade-off parameter $\alpha$, the number of approximation steps $T$, and the use of spectral normalization. We analyze them in the following, showing that Sm leads to enhanced results almost under any choice. This supports that our hypothesis — that neighboring instances are likely to have the same label — is a powerful inductive bias worth exploring.

Table 2: Classification results (mean and standard deviation from five independent runs). The best is in bold and the second-best is underlined. ($\downarrow$)/($\uparrow$) means lower/higher is better. The models with the proposed operator achieve the best performance overall, ranking first or second in nine out of twelve dataset-score pairs.

| | | RSNA | | PANDA | | CAMELYON16 | | |
| --- | --- | --- | --- | --- | --- | --- | --- | --- |
| | | AUROC ($\uparrow$) | F1 ($\uparrow$) | AUROC ($\uparrow$) | F1 ($\uparrow$) | AUROC ($\uparrow$) | F1 ($\uparrow$) | Rank ($\downarrow$) |
| Without global interactions | SmAP | $0.888_{0.005}$ | $\underline{0.787}_{0.026}$ | $\mathbf{0.943}_{0.001}$ | $\mathbf{0.915}_{0.002}$ | $\underline{0.976}_{0.007}$ | $\underline{0.916}_{0.016}$ | $\mathbf{1.833}_{0.753}$ |
| | ABMIL | $\underline{0.889}_{0.005}$ | $\mathbf{0.796}_{0.011}$ | $0.933_{0.002}$ | $\underline{0.909}_{0.001}$ | $0.956_{0.011}$ | $0.914_{0.021}$ | $2.500_{1.049}$ |
| | CLAM | $0.674_{0.157}$ | $0.161_{0.291}$ | $0.893_{0.026}$ | $0.868_{0.034}$ | $0.960_{0.029}$ | $0.897_{0.012}$ | $4.500_{0.837}$ |
| | DSMIL | $0.689_{0.063}$ | $0.240_{0.012}$ | $0.921_{0.008}$ | $0.904_{0.008}$ | $0.947_{0.076}$ | $0.866_{0.123}$ | $4.167_{0.753}$ |
| | DFTD-MIL | $\mathbf{0.890}_{0.045}$ | $0.775_{0.282}$ | $\underline{0.940}_{0.001}$ | $0.903_{0.002}$ | $\mathbf{0.983}_{0.01}$ | $\mathbf{0.937}_{0.013}$ | $\underline{2.000}_{1.265}$ |
| With global interactions | SmTAP | $\mathbf{0.906}_{0.007}$ | $\mathbf{0.825}_{0.026}$ | $0.946_{0.003}$ | $0.917_{0.002}$ | $\underline{0.976}_{0.014}$ | $\mathbf{0.948}_{0.02}$ | $\mathbf{1.833}_{0.983}$ |
| | TransMIL | $0.883_{0.008}$ | $0.716_{0.031}$ | $0.933_{0.010}$ | $0.895_{0.029}$ | $0.973_{0.018}$ | $0.911_{0.028}$ | $4.083_{0.917}$ |
| | SETMIL | $0.869_{0.011}$ | $0.716_{0.036}$ | $\mathbf{0.974}_{0.003}$ | $\mathbf{0.946}_{0.003}$ | $0.715_{0.155}$ | $0.471_{0.341}$ | $3.583_{2.010}$ |
| | GTP | $\underline{0.901}_{0.008}$ | $\underline{0.805}_{0.017}$ | $\underline{0.949}_{0.004}$ | $\underline{0.920}_{0.003}$ | $0.748_{0.118}$ | $0.727_{0.143}$ | $2.750_{0.987}$ |
| | CAMIL | $0.889_{0.019}$ | $0.805_{0.028}$ | $0.938_{0.003}$ | $0.911_{0.004}$ | $\mathbf{0.984}_{0.007}$ | $\underline{0.918}_{0.018}$ | $\underline{2.750}_{1.173}$ |

Table 3: Ablation study on different configurations of our models. AUROC (at both instance and bag levels), and normalized Dirichlet energy of attention values are reported. Almost all configurations improve the results in both tasks against the baseline (not using Sm).

| | RSNA | | | PANDA | | | CAMELYON16 | | |
| --- | --- | --- | --- | --- | --- | --- | --- | --- | --- |
| | AUROC$_{Inst}$($\uparrow$) | AUROC$_{Bag}$($\uparrow$) | $\mathcal{E}_D$ (**f**) | AUROC$_{Inst}$($\uparrow$) | AUROC$_{Bag}$($\uparrow$) | $\mathcal{E}_D$ (**f**) | AUROC$_{Inst}$($\uparrow$) | AUROC$_{Bag}$($\uparrow$) | $\mathcal{E}_D$ (**f**) |
| SmAP-early | 0.798 | 0.888 | 0.009 | 0.799 | 0.943 | 0.106 | 0.960 | 0.976 | 0.395 |
| SmAP-mid | 0.806 | 0.888 | 0.012 | 0.792 | 0.940 | 0.135 | 0.922 | 0.964 | 0.384 |
| SmAP-late | 0.811 | 0.891 | 0.011 | 0.802 | 0.944 | 0.082 | 0.819 | 0.964 | 0.321 |
| ABMIL | 0.806 | 0.889 | 0.023 | 0.768 | 0.933 | 0.141 | 0.819 | 0.956 | 0.419 |
| SmT+SmAP | 0.791 | 0.910 | 0.010 | 0.813 | 0.944 | 0.306 | 0.841 | 0.986 | 0.313 |
| SmT+AP | 0.791 | 0.910 | 0.010 | 0.754 | 0.940 | 0.356 | 0.754 | 0.984 | 0.320 |
| T+SmAP | 0.792 | 0.910 | 0.010 | 0.787 | 0.944 | 0.332 | 0.915 | 0.986 | 0.343 |
| T+AP | 0.792 | 0.910 | 0.020 | 0.760 | 0.942 | 0.391 | 0.781 | 0.984 | 0.433 |

### 5.3.1 Placement of Sm

Recall that SmAP leverages by default the *early* variation, but we also described SmAP-mid and SmAP-late. Likewise, we discussed different variants for SmTAP. Table 3 summarizes the impact of these choices on the final performance.

Sm **without the transformer encoder (**Sm**AP).** These variants differ in the place where Sm is located inside the attention pool, recall Eq. 10–Eq. 12. We include ABMIL since we build our model on top of it. We see that using SmAP improves the performance at both instance and bag levels. This improvement is more noticeable in PANDA and CAMELYON. We attribute it to the bag graph structure being more complex in WSIs than in CT scans. Also, the Dirichlet energy is lower when the smooth operator is used, as theoretically expected. We observe that the proposed method is robust to different placement configurations, which is consistent with the theoretical guarantees presented in Sec. 4.1. However, none of the variants consistently outperforms the others.

Sm **with the transformer encoder (**Sm**TAP).** Recall that SmTAP leverages Sm both after the transformer encoder and inside the attention pooling. Here we will refer to it as SmT+SmAP, and will compare it against T+SmAP and SmT+AP (using Sm only in one of the components) and against T+AP (not using Sm). We observe that Sm has no significant effect on bag-level performance. At instance-level we do observe differences: the baseline T+AP is outperformed as long as Sm is used within the attention pooling.

### 5.3.2 Sm **hyperparameters**

In the following we study the influence of the trade-off parameter $\alpha$ and of the spectral normalization. Due to space limitations, the analysis for the number of approximation steps $T$ is in Sec. B.3.

**The trade-off parameter $\alpha$.** From Eq. 6 we see that $\alpha \in [0, 1)$ controls the *amount of smoothness* enforced by Sm. Note that $\alpha = 0$ in Eq. 7 produces no smoothness, turning Sm into the identity operator. In Fig. 6a we show the performance obtained for different values of $\alpha$ in CAMELYON16. Each choice of this hyperparameter improves upon the baseline ABMIL ($\alpha = 0$). We see that better localization results are obtained when $\alpha$ is lower, while better classification results are obtained when

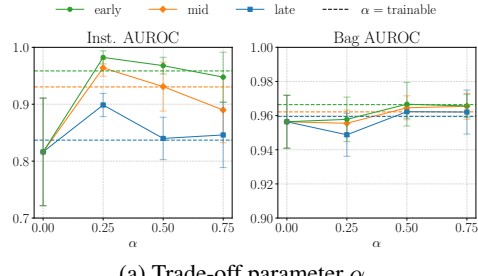
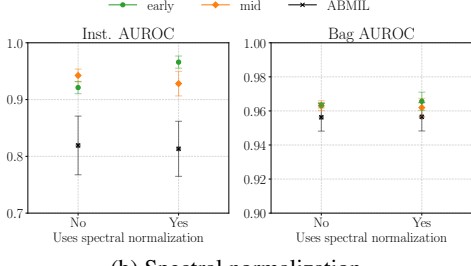

(a) Trade-off parameter $\alpha$.  (b) Spectral normalization.

Figure 6: Influence of the trade-off parameter $\alpha$ (left) and of spectral normalization (right) in CAMELYON16. Setting $\alpha > 0$ improves upon the baseline ABMIL ($\alpha = 0$) and is a trade-off between better localization results (lower $\alpha$) or better classification results (higher $\alpha$). Likewise, Sm without spectral normalization already improves the results upon the baseline (ABMIL), but the best performance is obtained when they are used together.

$\alpha$ is higher. Fixing $\alpha = 0.5$ is a compromise between the two, and produces very similar results as setting it as a trainable parameter initialized at $\alpha = 0.5$. Fig. 12 provides a visual comparison of the effect that $\alpha$ has on the attention maps.

**The effect of spectral normalization.** Spectral normalization forces the norm of the multi-layer perceptron weights to remain constant. In this work, this is a key design choice that helps Sm to obtain attention maps with lower Dirichlet energy. In our experiments, we have used spectral normalization in the layers immediately after Sm. Note that the late variant does not require spectral normalization, since it applies Sm directly to the attention values. In Fig. 6b we show the results obtained with and without spectral normalization in CAMELYON16. We observe that, even without spectral normalization, Sm improves upon the baseline. The improvement is more significant when Sm is paired with spectral normalization, especially at the instance level.

## 6 Discussion and conclusion

The main goal of this paper is to draw attention to the study of MIL methods at the instance level. To that end, we revised current deep MIL methods and provided a unified perspective on them. We proposed the smooth operator Sm to introduce local interactions in a principled way. By design, it produces smooth attention maps that resemble the ground truth instance labels. We conducted an exhaustive experimental validation with three real-world MIL datasets and up to eight state-of-the-art methods in both classification and localization tasks. This study showed that our method provides the best performance in localization while being highly competitive (best or second best) at classification.

Despite its advantages, our method has some limitations. The first is that, as with every other operator in GNNs, the computational costs of the smooth operator scale with the size of the bag. Fortunately, it can be paired with existing subgraph sampling techniques to mitigate this problem. The second limitation is that we do not have a definite answer for where it is better to use the proposed operator. We have shown that it leads to improvements in almost every place, but the optimal location may be problem-dependent and has to be tailored by the practitioner.

Finally, we hope that our work will draw more attention to the localization problem, which is very important for the deployment of computer-aided systems in the real world. In this sense, safely deploying the proposed methods in clinical practice requires evaluating them in a wider range of medical problems and quantifying their uncertainty. For the latter, we believe that the smooth operator could also benefit from a probabilistic formulation.

## Acknowledgements

This work was supported by project PID2022-140189OB-C22 funded by MCIN / AEI / 10.13039 / 501100011033. Francisco M. Castro-Macías acknowledges FPU contract FPU21/01874 funded by Ministerio de Universidades. Pablo Morales-Álvarez acknowledges grant C-EXP-153-UGR23 funded by Consejería de Universidad, Investigación e Innovación and by the European Union (EU) ERDF Andalusia Program 2021-2027.

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

# A Proofs

## A.1 Proof of Eq. 4

We present a general result for an arbitrary multilayer perceptron with Lipschitz activation functions. Note that assuming Lipschitzness is not a restriction, since most of the currently used activation functions meet this property [6]. The desired Eq. 4 is a particular case of this result. Let $L \in \mathbb{N}$. Consider a $L$-layer perceptron that, given $\mathbf{Y} \in \mathbb{R}^{N \times D_0}$, outputs $\mathbf{Y}^L \in \mathbb{R}^{N \times D_L}$ defined by the following rule

$$\mathbf{Y}^0 = \mathbf{Y}, \tag{14}$$

$$\mathbf{Y}^{\ell+1} = \varphi_\ell \left( \mathbf{Y}^\ell \mathbf{W}_\ell + \mathbf{B}_\ell \right), \quad \ell \in \{0, \dots, L-1\}, \tag{15}$$

where $\mathbf{W}_\ell \in \mathbb{R}^{D_\ell \times D_{\ell+1}}$, and $\mathbf{B}_\ell = [\mathbf{b}_\ell, \dots, \mathbf{b}_\ell]^\top \in \mathbb{R}^{N \times D_{\ell+1}}$ where $\mathbf{b}_\ell \in \mathbb{R}^{D_{\ell+1}}$ are trainable weights, and $\varphi_\ell \colon \mathbb{R} \to \mathbb{R}$ are activation functions applied element-wise. We suppose that each activation function $\varphi_\ell$ is $K_\ell$-Lipschitz. Then, we obtain the following inequality,

$$\mathcal{E}_D \left( \mathbf{Y}^{\ell+1} \right) \leq K_\ell^2 \left\| \mathbf{W}_\ell \right\|_2^2 \mathcal{E}_D \left( \mathbf{Y}^\ell \right). \tag{16}$$

Before verifying it, we note that by applying this inequality to every layer, we arrive at

$$\mathcal{E}_D \left( \mathbf{Y}^L \right) \leq \cdots \leq K_{L-1-\ell:0}^2 \left\| \mathbf{W}_{L-1-\ell:0} \right\|^2 \mathcal{E}_D \left( \mathbf{Y}^\ell \right) \leq \cdots \leq K_{L-1:0}^2 \left\| \mathbf{W}_{L-1:0} \right\|^2 \mathcal{E}_D \left( \mathbf{Y} \right), \tag{17}$$

where $\left\| \mathbf{W}_{\ell:0} \right\|^2 = \prod_{j=0}^\ell \left\| \mathbf{W}_j \right\|^2$ and $K_{\ell:0} = \prod_{j=0}^\ell K_\ell$. Taking $L = 2$, $D_0 = D$, $D_1 = L$, $D_2 = 1$, $\mathbf{b}_0 = \mathbf{b}_1 = \mathbf{0}$, $\varphi_0 = \tanh$, and $\varphi_1 = \text{Id}$, we recover Eq. 4.

To verify Eq. 16, we write $\mathbf{Y}^{\ell+1} = \left[ \mathbf{y}_1^{\ell+1}, \dots, \mathbf{y}_N^{\ell+1} \right]^\top$ and $\mathbf{Y}^\ell = \left[ \mathbf{y}_1^\ell, \dots, \mathbf{y}_N^\ell \right]^\top$. We have,

$$\mathcal{E}_D \left( \mathbf{Y}^{\ell+1} \right) = \frac{1}{2} \sum_{i=1}^N \sum_{j=1}^N A_{ij} \left\| \mathbf{y}_i^{\ell+1} - \mathbf{y}_j^{\ell+1} \right\|_2^2 = \tag{18}$$

$$= \frac{1}{2} \sum_{i=1}^N \sum_{j=1}^N A_{ij} \left\| \varphi_\ell \left( \mathbf{W}_\ell^\top \mathbf{y}_i^\ell + \mathbf{B}_\ell \right) - \varphi_\ell \left( \mathbf{W}_\ell^\top \mathbf{y}_j^\ell + \mathbf{B}_\ell \right) \right\|_2^2 \leq \tag{19}$$

$$\leq K_\ell^2 \frac{1}{2} \sum_{i=1}^N \sum_{j=1}^N A_{ij} \left\| \mathbf{W}_\ell^\top \left( \mathbf{y}_i^\ell - \mathbf{y}_j^\ell \right) \right\|_2^2 \leq \tag{20}$$

$$\leq K_\ell^2 \left\| \mathbf{W}_\ell \right\|_2^2 \frac{1}{2} \sum_{i=1}^N \sum_{j=1}^N A_{ij} \left\| \mathbf{y}_i^\ell - \mathbf{y}_j^\ell \right\|_2^2 = K_\ell^2 \left\| \mathbf{W}_\ell \right\|_2^2 \mathcal{E}_D \left( \mathbf{Y}^\ell \right), \tag{21}$$

where from Eq. 19 to Eq. 20 we have used the definition of Lipschitz function and from Eq. 20 to Eq. 21 we have used the consistency between the spectral and Euclidean norms.

## A.2 Proof of Eq. 9

In this section, we adapt the proof presented in [6] for a similar result. Let $\mathbf{U} \in \mathbb{R}^{N \times D}$. Our goal is to show that

$$\mathcal{E}_D \left( (\mathbf{I} + \gamma \mathbf{L})^{-1} \mathbf{U} \right) \leq \lambda_\gamma^* \mathcal{E}_D \left( \mathbf{U} \right), \tag{22}$$

where $\gamma > 0$ and $\lambda_\gamma^* = \max \left\{ (1 + \gamma \lambda_n)^{-2} : \lambda_n \in \Lambda \setminus \{0\} \right\}$, being $\Lambda = \{\lambda_1, \dots, \lambda_N\}$ the eigenvalues of the symmetric graph Laplacian matrix $\mathbf{L}$. First, we reduce the proof to univariate graph functions by looking at the rows of $\mathbf{U}$ as univariate graph functions. Denoting them as $\{\mathbf{u}_1, \dots, \mathbf{u}_D\}$, where each $\mathbf{u}_d \in \mathbb{R}^N$, we have $\mathcal{E}_D \left( (\mathbf{I} + \gamma \mathbf{L})^{-1} \mathbf{U} \right) = \sum_{d=1}^D \mathcal{E}_D \left( (\mathbf{I} + \gamma \mathbf{L})^{-1} \mathbf{u}_d \right)$. Therefore, it will be sufficient to show that, for any $\mathbf{u} \in \mathbb{R}^N$,

$$\mathcal{E}_D \left( (\mathbf{I} + \gamma \mathbf{L})^{-1} \mathbf{u} \right) \leq \lambda_\gamma^* \mathcal{E}_D \left( \mathbf{u} \right). \tag{23}$$

Next, it is useful to note that if $\lambda_n$ is an eigenvalue of $\mathbf{L}$ with associated eigenvector $\mathbf{v}_n$, then $(1 + \gamma\lambda_n)^{-1}$ is an eigenvalue of $(\mathbf{I} + \gamma\mathbf{L})^{-1}$ with the same associated eigenvector. Finally, let $\{\mathbf{v}_1, \ldots, \mathbf{v}_N\}$ be a orthonormal eigenbasis of $\mathbf{L}$, being each $\mathbf{v}_n$ associated to the eigenvalue $\lambda_n$. This basis always exists since $\mathbf{L}$ is a symmetric matrix. Writing $\mathbf{u} = \sum_{n=1}^{N} c_n\mathbf{v}_n$, with $c_n \in \mathbb{R}$, we have

$$(\mathbf{I} + \gamma\mathbf{L})^{-1}\mathbf{u} = \sum_{n=1}^{N} c_n (1 + \gamma\lambda_n)^{-1}\mathbf{v}_n. \tag{24}$$

Using that the eigenvectors are orthogonal to each other, we arrive at

$$\mathcal{E}_D\left((\mathbf{I} + \gamma\mathbf{L})^{-1}\mathbf{u}\right) = \sum_{n=1}^{N} c_n^2 \lambda_n (1 + \gamma\lambda_n)^{-2} \leq \lambda_\gamma^* \sum_{n=1}^{N} c_n^2 \lambda_n = \lambda_\gamma^* \mathcal{E}_D(\mathbf{u}). \tag{25}$$

## B   Experiments: details and further results

In this section, we provide the details of the datasets, architectures, and configurations used for each experiment. The code is available at `https://github.com/Franblueee/SmMIL`.

### B.1   Datasets

We provide insights into the datasets we have used: a description of the problem, the train/test splits, and preprocessing (instance selection and feature extraction). For all datasets, we obtain an initial train/test partition. Then, we split the initial train partition into five different train/validation splits. Every model is trained on each of these splits and then evaluated on the test set. We report the average performance on this test set.

**RSNA.** It was published by the Radiological Society of North America (RSNA) to detect acute intracranial hemorrhage and its subtypes [13]. It is available in Kaggle[1]. We use the official train-test split. It includes a total of 1150 scans. There are a total amount of 39750 slices and the number in each scan varies from 24 to 57. Each slice is preprocessed following [36].

**PANDA.** It is a public dataset for the classification of the severity of prostate cancer from microscopy scans of prostate biopsy samples [5]. It is available in Kaggle[2]. Since the official test set is not publicly available, we use the train/test split proposed in [33]. To extract the patches from each WSI, we follow the procedure described in [33], obtaining patches of size $512 \times 512$ at $10\times$ magnification. This results in a total amount of 10503 WSIs and 1107931 patches.

**CAMELYON16.** It is a public dataset for the detection of breast cancer metastasis [2]. It is available at the Registry of Open Data of AWS[3]. The official repository contains 400 WSIs in total, including 270 for training and 130 for testing. From each WSI, we extract patches of size $512 \times 512$ at $20\times$ magnification using the method proposed by Lu et al. [25].

### B.2   Model and training configuration

We provide details about how we have implemented the proposed methods and how we have conducted the experiments.

**Feature extractor.** Due to the limited memory of the GPU, it is necessary to extract features from each instance. Otherwise, the bags will not fit in memory. In this work, we consider three options for the feature extractor, all of which are pre-trained in Imagenet: ResNet18 ($P = 512$), ResNet50 ($P = 2048$), and ViT-B-32 ($P = 768$). In addition, for CAMELYON16 we also consider ResNet50-BT[4] ($P = 2048$), which is a ResNet50 model pre-trained using the Barlow Twins Self-Supervised Learning method on a huge dataset of WSIs patches [38, 18]. The results reported in the main text correspond to ResNet18 for RSNA and PANDA, and to ResNet50-BT for CAMELYON16. We study how the choice of the feature extractor affects the results in Sec. B.3.

---

[1]`https://www.kaggle.com/c/rsna-intracranial-hemorrhage-detection`
[2]`https://www.kaggle.com/c/prostate-cancer-grade-assessment/data`
[3]`https://registry.opendata.aws/camelyon/`
[4]Weights available at `https://github.com/lunit-io/benchmark-ssl-pathology`.

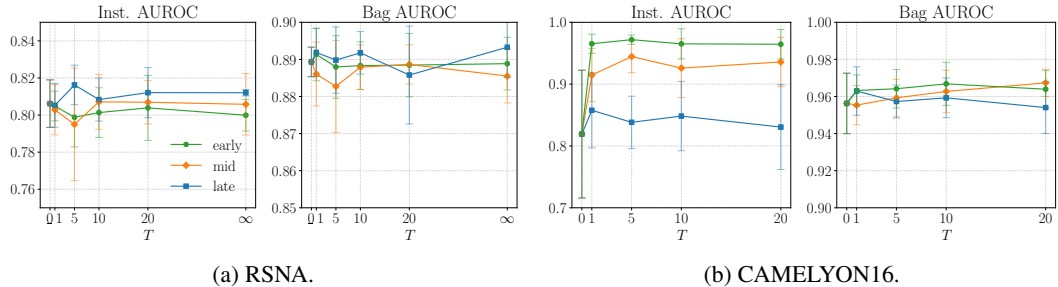

(a) RSNA.  (b) CAMELYON16.

Figure 7: Influence of the number of steps $T$ used to approximate Sm in RSNA and CAMELYON16. ABMIL corresponds to $T = 0$. Using $T = 10$ is enough to closely match the performance of the exact form ($T = \infty$).

**Model architecture.** We describe the architecture we have used for the proposed methods (SmAP and SmTAP). For the rest of the methods considered, we adopt their original implementations and default configurations, publicly available on their GitHub repositories. For the independent instance encoding part (see Fig. 3a and Fig. 3b), the instance embeddings $\mathbf{h}_n$ are obtained using one fully connected layer with 512 units ($D = 512$). For the attention-based pooling described by Eq. 1 and Eq. 2, we fix $D = 512$ and $L = 100$. The transformer encoder in Fig. 3c is implemented using two transformer layers. These layers use the standard multi-head attention mechanism equipped with skip connections and layer normalization [4]. We fix the key, query, and value dimensions to 128 and the number of heads to 8. We used the Pytorch's implementation of dot product attention[5]. Finally, the bag-embedding classifier was implemented using one fully connected layer.

**Training setup and hyperparameters.** To ensure fair and reproducible conditions, we trained every method under the same setup. The number of epochs was set to 50. We adopt the Adam optimizer [19] with the default Pytorch configuration. For the base learning rate, we considered two different values, $10^{-4}$ and $10^{-5}$, since we noticed that models that do not use transformers obtained better results when the learning rate was higher. We report the best results for each model. We adopted a slow start using Pytorch's `LinearLR` scheduler with `start_factor=0.1` and `total_iters=5`. During training, we monitored the bag AUROC and cross-entropy loss in the validation set and kept the weights that obtained the best results. The batch size was set to 32 in RSNA and PANDA. In CAMELYON16, it was set to 4 for no-transformer methods, and to 1 for transformer-based methods. However, for SETMIL, we had to set it to 1 in PANDA and CAMELYON16 due to its high GPU memory requirements. In RSNA we weighted the loss function to account for the imbalance between positive and negative bags since we observed it to improve the results. All the experiments were performed on one NVIDIA GeForce RTX 3090.

### B.3 Further ablation studies.

We complete the ablation study presented in the main paper, Sec. 5.3, by looking at the rest of the design choices or hyperparameters associated with our Sm.

**Smooth operator approximation.** The exact form of Sm given by Eq. 7 becomes computationally infeasible for large bag sizes. The quality of the approximation, given by Eq. 8, is controlled by the number of steps $T$. Fig. 7 shows the results for different values of this hyperparameter in RSNA and CAMELYON16. In RSNA, since the bags are smaller, we can compute the closed-form solution, which we represent as $T = \infty$. Almost any choice of $T > 0$ improves upon ABMIL. This improvement is particularly noticeable in CAMELYON16. Moreover, in most cases, the performance stabilizes at $T = 10$, which is the value we used in our experiments.

**Sm on top of other models.** We have proposed two new models (SmAP and SmTAP) by applying Sm on top of two baselines (ABMIL and Transformer+ABMIL, respectively). Instead, the Sm can be applied on top of other existing approaches. In Table 4 we explore how other approaches behave when combined with the proposed Sm in the CAMELYON16 dataset. Instance-level performance is enhanced (greatly in some cases, e.g. an increase from 0.76 to 0.96 in AUROC for DSMIL), whereas bag-level results are competitive. The decrease in bag-level results for DFTD-MIL is explained by

---
[5]`https://pytorch.org/docs/stable/generated/torch.nn.functional.scaled_dot_product_attention.html`

Table 4: Using Sm on top of other models (CAMELYON16 with ResNet50-BT features). Improvements are highlighted in green. Using the proposed Sm increases the instance-level performance, while the bag-level performance remains competitive.

| | Instance | | Bag | |
|---|---|---|---|---|
| | AUROC ($\uparrow$) | F1 ($\uparrow$) | AUROC ($\uparrow$) | F1 ($\uparrow$) |
| CLAM | $0.849_{0.044}$ | $0.821_{0.046}$ | $0.96_{0.029}$ | $0.897_{0.012}$ |
| SmCLAM | $0.928_{0.028}$ | $0.873_{0.018}$ | $0.966_{0.007}$ | $0.889_{0.017}$ |
| DSMIL | $0.76_{0.078}$ | $0.654_{0.203}$ | $0.947_{0.085}$ | $0.866_{0.136}$ |
| SmDSMIL | $0.960_{0.013}$ | $0.776_{0.088}$ | $0.967_{0.011}$ | $0.919_{0.018}$ |
| DFTD-MIL | $0.984_{0.002}$ | $0.742_{0.040}$ | $0.983_{0.010}$ | $0.937_{0.013}$ |
| SmDFTD-MIL | $0.984_{0.183}$ | $0.836_{0.222}$ | $0.978_{0.158}$ | $0.903_{0.183}$ |

Table 5: Instance and bag AUROC (higher is better) in CAMELYON16 using ResNet50-BT features for the proposed methods and the penalty-based approach. The best in each column is highlighted in bold. Sm obtains superior performance, although the differences are not large.

| | | RSNA | | PANDA | | CAMELYON16 | |
|---|---|---|---|---|---|---|---|
| | | Inst. | Bag | Inst. | Bag | Inst. | Bag |
| W/o global int. | SmAP | $\mathbf{0.798}_{0.033}$ | $0.888_{0.005}$ | $\mathbf{0.799}_{0.005}$ | $\mathbf{0.943}_{0.001}$ | $0.961_{0.007}$ | $\mathbf{0.965}_{0.007}$ |
| | ABMIL+PENALTY | $0.782_{0.050}$ | $\mathbf{0.889}_{0.043}$ | $0.780_{0.003}$ | $0.935_{0.001}$ | $\mathbf{0.979}_{0.013}$ | $0.963_{0.012}$ |
| W/ global int. | SmTAP | $\mathbf{0.767}_{0.046}$ | $\mathbf{0.906}_{0.007}$ | $\mathbf{0.790}_{0.007}$ | $0.946_{0.003}$ | $\mathbf{0.789}_{0.008}$ | $0.976_{0.014}$ |
| | T+PENALTY | $0.737_{0.045}$ | $0.905_{0.005}$ | $0.772_{0.011}$ | $\mathbf{0.947}_{0.001}$ | $0.769_{0.099}$ | $\mathbf{0.988}_{0.004}$ |

the fact that this method randomly splits each bag into different chunks. This may lead to the loss of local interactions exploited by Sm (e.g. if two adjacent instances end in different chunks).

**An alternative smoothing strategy.** Introducing a penalty term in the loss function to favor smoothness is a natural alternative to the proposed operator. However, there is an important difference: the use of a penalty term does not modify the model architecture. The penalty term favors that the learned weights encode such a property, but it is not explicitly encoded in the model. For instance, note that the penalty term is not used at inference time. We compare the penalty-based approach and the proposed Sm in Table 5. Although differences are not large, Sm obtains superior performance.

**Feature extractor.** We investigate whether the choice of the feature extractor influences the results and conclusions presented in the main text. We have evaluated each of the considered methods in each dataset using the feature extractors mentioned above (ResNet18, ResNet50, ViT-B-32, and ResNet50-BT). The results are shown in Tables 7–10. We summarize them in Table 6, where we collect the average instance and bag rank of each method for each feature extractor. We observe that the proposed smooth operator Sm obtains in almost all cases the highest rank. This supports the idea that the improvement introduced by Sm does not depend on the used features.

Table 6: Instance and bag average ranks (lower is better) obtained by each method for different choices of the feature extractor. The best result within each group is bolded, and the second-best is underlined. SmAP and SmTAP obtain in almost all cases the highest rank.

| | | ResNet18 | | ResNet50 | | ViT-B-32 | | ResNet50-BT | |
|---|---|---|---|---|---|---|---|---|---|
| | | Inst. | Bag | Inst. | Bag | Inst. | Bag | Inst. | Bag |
| Without global interactions | SmAP | $\mathbf{2.000}_{0.632}$ | $\underline{2.000}_{1.095}$ | $\mathbf{1.625}_{0.744}$ | $1.750_{0.707}$ | $\mathbf{1.667}_{0.816}$ | $\mathbf{1.500}_{1.225}$ | $\mathbf{1.000}_{0.000}$ | $\underline{2.000}_{0.000}$ |
| | ABMIL | $2.667_{1.366}$ | $\mathbf{1.667}_{0.816}$ | $3.750_{1.581}$ | $3.250_{0.707}$ | $4.333_{2.160}$ | $3.000_{0.894}$ | $4.500_{0.707}$ | $3.500_{0.707}$ |
| | DeepGraphSurv | $4.000_{2.000}$ | $5.500_{1.225}$ | $\underline{2.500}_{1.195}$ | $5.625_{0.916}$ | $3.333_{1.211}$ | $5.000_{0.000}$ | $\underline{2.500}_{0.707}$ | $6.000_{0.000}$ |
| | CLAM | $6.167_{0.983}$ | $5.500_{1.225}$ | $4.750_{2.053}$ | $4.500_{2.070}$ | $6.333_{0.816}$ | $5.000_{2.449}$ | $3.000_{1.414}$ | $3.500_{0.707}$ |
| | DSMIL | $5.167_{0.983}$ | $5.333_{1.506}$ | $6.375_{0.518}$ | $6.000_{0.926}$ | $5.667_{1.033}$ | $6.667_{0.516}$ | $6.000_{0.000}$ | $5.000_{0.000}$ |
| | PathGCN | $5.833_{1.472}$ | $5.167_{1.722}$ | $5.625_{1.302}$ | $4.625_{2.134}$ | $4.500_{1.643}$ | $4.000_{1.673}$ | $7.000_{0.000}$ | $7.000_{0.000}$ |
| | DFTD-MIL | $\underline{2.167}_{0.983}$ | $2.833_{1.169}$ | $3.375_{1.302}$ | $\underline{2.250}_{1.488}$ | $\underline{2.167}_{0.983}$ | $\underline{2.833}_{0.983}$ | $4.000_{1.414}$ | $\mathbf{1.000}_{0.000}$ |
| With global interactions | SmTAP | $\underline{2.167}_{1.835}$ | $\mathbf{1.667}_{1.033}$ | $\underline{2.375}_{1.847}$ | $\mathbf{1.875}_{0.835}$ | $\mathbf{1.833}_{0.983}$ | $2.500_{0.837}$ | $\underline{1.500}_{0.707}$ | $\mathbf{1.500}_{0.707}$ |
| | TransMIL | $3.167_{1.329}$ | $3.833_{1.169}$ | $3.500_{1.069}$ | $3.625_{1.061}$ | $4.167_{1.329}$ | $4.500_{1.975}$ | $4.000_{1.414}$ | $4.000_{0.000}$ |
| | SETMIL | $3.667_{0.816}$ | $3.500_{1.975}$ | $3.625_{1.768}$ | $4.125_{2.031}$ | $3.667_{1.506}$ | $3.333_{1.862}$ | $4.500_{0.707}$ | $6.000_{0.000}$ |
| | GTP | $4.167_{0.983}$ | $3.333_{1.751}$ | $4.500_{1.069}$ | $3.375_{1.598}$ | $5.000_{1.265}$ | $4.833_{1.329}$ | $6.000_{0.000}$ | $5.000_{0.000}$ |
| | IIBMIL | $5.167_{2.041}$ | $5.667_{0.816}$ | $4.500_{2.138}$ | $5.125_{1.642}$ | $4.167_{1.722}$ | $3.167_{2.041}$ | $\mathbf{2.000}_{1.414}$ | $2.500_{0.707}$ |
| | CAMIL | $\mathbf{2.667}_{1.751}$ | $\underline{3.000}_{0.894}$ | $\mathbf{2.500}_{1.309}$ | $\underline{2.875}_{1.356}$ | $\underline{2.167}_{1.472}$ | $\underline{2.667}_{1.211}$ | $3.000_{1.414}$ | $\underline{2.000}_{1.414}$ |

Table 7: Instance AUROC (higher is better) for different choices of the feature extractor.

| | | ResNet18 | | | ResNet50 | | | ViT-B-32 | | | ResNet50+BT |
|---|---|---|---|---|---|---|---|---|---|---|---|
| | | RSNA | PANDA | CAMELYON16 | RSNA | PANDA | CAMELYON16 | RSNA | PANDA | CAMELYON16 | CAMELYON16 |
| Without global interactions | SmAP | $0.798_{0.033}$ | $\mathbf{0.799}_{0.005}$ | $0.798_{0.037}$ | $0.783_{0.026}$ | $0.786_{0.005}$ | $0.854_{0.116}$ | $0.804_{0.017}$ | $\mathbf{0.810}_{0.007}$ | $0.773_{0.062}$ | $\mathbf{0.961}_{0.007}$ |
| | ABMIL | $\mathbf{0.806}_{0.012}$ | $0.768_{0.002}$ | $0.679_{0.082}$ | $\mathbf{0.796}_{0.027}$ | $0.774_{0.009}$ | $0.806_{0.130}$ | $0.797_{0.023}$ | $0.773_{0.004}$ | $0.755_{0.143}$ | $0.816_{0.055}$ |
| | DeepGraphSurv | $0.681_{0.054}$ | $0.720_{0.011}$ | $0.868_{0.094}$ | $0.768_{0.013}$ | $\mathbf{0.806}_{0.002}$ | $0.814_{0.027}$ | $0.755_{0.063}$ | $0.809_{0.008}$ | $0.756_{0.104}$ | $0.959_{0.033}$ |
| | CLAM | $0.523_{0.069}$ | $0.727_{0.046}$ | $0.516_{0.102}$ | $0.497_{0.005}$ | $0.785_{0.004}$ | $0.559_{0.056}$ | $0.500_{0.000}$ | $0.777_{0.004}$ | $0.463_{0.034}$ | $0.849_{0.044}$ |
| | DSMIL | $0.554_{0.004}$ | $0.765_{0.008}$ | $0.628_{0.181}$ | $0.568_{0.015}$ | $0.747_{0.006}$ | $0.670_{0.111}$ | $0.702_{0.029}$ | $0.779_{0.002}$ | $0.661_{0.115}$ | $0.760_{0.078}$ |
| | PathGCN | $0.711_{0.049}$ | $0.664_{0.019}$ | $0.618_{0.214}$ | $0.692_{0.047}$ | $0.772_{0.011}$ | $0.851_{0.219}$ | $0.749_{0.046}$ | $0.769_{0.032}$ | $0.813_{0.073}$ | $0.443_{0.138}$ |
| | DFTD-MIL | $0.747_{0.070}$ | $0.795_{0.004}$ | $\mathbf{0.920}_{0.074}$ | $0.795_{0.018}$ | $0.784_{0.007}$ | $\mathbf{0.952}_{0.013}$ | $\mathbf{0.807}_{0.030}$ | $0.785_{0.008}$ | $\mathbf{0.952}_{0.011}$ | $0.884_{0.002}$ |
| With global interactions | SmTAP | $\mathbf{0.767}_{0.046}$ | $\mathbf{0.790}_{0.007}$ | $0.750_{0.134}$ | $\mathbf{0.802}_{0.016}$ | $0.756_{0.012}$ | $0.716_{0.105}$ | $\mathbf{0.795}_{0.027}$ | $\mathbf{0.822}_{0.010}$ | $0.819_{0.162}$ | $0.789_{0.008}$ |
| | TransMIL | $0.732_{0.013}$ | $0.751_{0.011}$ | $0.820_{0.038}$ | $0.707_{0.023}$ | $0.743_{0.021}$ | $0.844_{0.023}$ | $0.749_{0.019}$ | $0.749_{0.040}$ | $0.779_{0.062}$ | $0.781_{0.024}$ |
| | SETMIL | $0.726_{0.025}$ | $0.774_{0.007}$ | $0.792_{0.032}$ | $0.678_{0.004}$ | $\mathbf{0.774}_{0.071}$ | $0.787_{0.005}$ | $0.755_{0.001}$ | $0.789_{0.089}$ | $0.569_{0.009}$ | $0.615_{0.231}$ |
| | GTP | $0.736_{0.017}$ | $0.768_{0.022}$ | $0.594_{0.228}$ | $0.736_{0.024}$ | $0.754_{0.019}$ | $0.528_{0.149}$ | $0.760_{0.013}$ | $0.720_{0.039}$ | $0.477_{0.095}$ | $0.442_{0.091}$ |
| | IIBMIL | $0.675_{0.017}$ | $0.740_{0.020}$ | $0.500_{0.000}$ | $0.672_{0.024}$ | $0.729_{0.031}$ | $0.500_{0.000}$ | $0.690_{0.017}$ | $0.740_{0.031}$ | $0.863_{0.038}$ | $\mathbf{0.873}_{0.138}$ |
| | CAMIL | $0.760_{0.036}$ | $0.785_{0.011}$ | $\mathbf{0.917}_{0.043}$ | $0.796_{0.013}$ | $0.766_{0.027}$ | $\mathbf{0.964}_{0.014}$ | $0.783_{0.007}$ | $0.806_{0.015}$ | $\mathbf{0.939}_{0.007}$ | $0.742_{0.028}$ |

Table 8: Instance F1 (higher is better) for different choices of the feature extractor.

| | | ResNet18 | | | ResNet50 | | | ViT-B-32 | | | ResNet50+BT |
|---|---|---|---|---|---|---|---|---|---|---|---|
| | | RSNA | PANDA | CAMELYON16 | RSNA | PANDA | CAMELYON16 | RSNA | PANDA | CAMELYON16 | CAMELYON16 |
| Without global interactions | SmAP | $0.477_{0.014}$ | $0.635_{0.006}$ | $0.591_{0.059}$ | $\mathbf{0.473}_{0.015}$ | $0.630_{0.009}$ | $\mathbf{0.675}_{0.077}$ | $0.494_{0.019}$ | $\mathbf{0.645}_{0.009}$ | $\mathbf{0.580}_{0.053}$ | $\mathbf{0.839}_{0.053}$ |
| | ABMIL | $\mathbf{0.486}_{0.033}$ | $0.602_{0.004}$ | $0.428_{0.049}$ | $0.470_{0.031}$ | $0.611_{0.007}$ | $0.654_{0.067}$ | $\mathbf{0.498}_{0.021}$ | $0.605_{0.006}$ | $0.419_{0.029}$ | $0.767_{0.039}$ |
| | DeepGraphSurv | $0.293_{0.168}$ | $0.581_{0.026}$ | $\mathbf{0.595}_{0.129}$ | $0.464_{0.022}$ | $\mathbf{0.641}_{0.002}$ | $0.663_{0.038}$ | $0.479_{0.043}$ | $0.642_{0.006}$ | $0.465_{0.059}$ | $0.771_{0.070}$ |
| | CLAM | $0.076_{0.154}$ | $0.568_{0.038}$ | $0.406_{0.238}$ | $0.000_{0.000}$ | $0.621_{0.007}$ | $0.584_{0.087}$ | $0.000_{0.000}$ | $0.610_{0.005}$ | $0.373_{0.054}$ | $0.821_{0.046}$ |
| | DSMIL | $0.180_{0.000}$ | $0.598_{0.006}$ | $0.155_{0.180}$ | $0.271_{0.019}$ | $0.592_{0.005}$ | $0.290_{0.174}$ | $0.399_{0.031}$ | $0.610_{0.004}$ | $0.255_{0.134}$ | $0.654_{0.203}$ |
| | PathGCN | $0.447_{0.014}$ | $0.526_{0.019}$ | $0.150_{0.211}$ | $0.431_{0.020}$ | $0.608_{0.010}$ | $0.371_{0.211}$ | $0.481_{0.039}$ | $0.610_{0.023}$ | $0.414_{0.119}$ | $0.077_{0.114}$ |
| | DFTD-MIL | $0.453_{0.194}$ | $\mathbf{0.637}_{0.006}$ | $0.563_{0.132}$ | $0.447_{0.026}$ | $0.617_{0.011}$ | $0.591_{0.098}$ | $0.489_{0.033}$ | $0.616_{0.013}$ | $0.552_{0.055}$ | $0.742_{0.040}$ |
| With global interactions | SmTAP | $\mathbf{0.474}_{0.023}$ | $0.622_{0.010}$ | $\mathbf{0.581}_{0.061}$ | $\mathbf{0.517}_{0.020}$ | $0.606_{0.015}$ | $\mathbf{0.630}_{0.070}$ | $0.475_{0.034}$ | $0.655_{0.013}$ | $\mathbf{0.552}_{0.138}$ | $\mathbf{0.600}_{0.067}$ |
| | TransMIL | $0.471_{0.014}$ | $0.636_{0.008}$ | $0.174_{0.080}$ | $0.442_{0.024}$ | $0.622_{0.023}$ | $0.196_{0.115}$ | $0.480_{0.046}$ | $0.630_{0.041}$ | $0.194_{0.090}$ | $0.127_{0.078}$ |
| | SETMIL | $0.438_{0.027}$ | $0.631_{0.010}$ | $0.237_{0.058}$ | $0.405_{0.021}$ | $\mathbf{0.821}_{0.022}$ | $0.036_{0.021}$ | $0.467_{0.008}$ | $\mathbf{0.822}_{0.012}$ | $0.159_{0.039}$ | $0.134_{0.267}$ |
| | GTP | $0.425_{0.018}$ | $0.636_{0.011}$ | $0.168_{0.132}$ | $0.431_{0.013}$ | $0.621_{0.014}$ | $0.150_{0.122}$ | $0.447_{0.021}$ | $0.641_{0.017}$ | $0.084_{0.048}$ | $0.037_{0.036}$ |
| | IIBMIL | $0.420_{0.016}$ | $\mathbf{0.645}_{0.007}$ | $0.000_{0.000}$ | $0.403_{0.014}$ | $0.641_{0.019}$ | $0.000_{0.000}$ | $0.443_{0.010}$ | $0.655_{0.006}$ | $0.295_{0.015}$ | $0.352_{0.100}$ |
| | CAMIL | $0.456_{0.013}$ | $0.621_{0.013}$ | $0.403_{0.157}$ | $0.483_{0.024}$ | $0.615_{0.014}$ | $0.563_{0.153}$ | $\mathbf{0.504}_{0.025}$ | $0.641_{0.014}$ | $0.426_{0.055}$ | $0.479_{0.175}$ |

Table 9: Bag AUROC (higher is better) for different choices of the feature extractor.

| | | ResNet18 | | | ResNet50 | | | ViT-B-32 | | ResNet50+BT |
| | | RSNA | PANDA | CAMELYON16 | RSNA | PANDA | CAMELYON16 | RSNA | PANDA | CAMELYON16 | CAMELYON16 |
|---|---|---|---|---|---|---|---|---|---|---|---|
| Without global interactions | SmAP | $0.888_{0.005}$ | $\mathbf{0.943}_{0.001}$ | $0.729_{0.037}$ | $\mathbf{0.890}_{0.007}$ | $0.944_{0.001}$ | $\mathbf{0.777}_{0.046}$ | $\mathbf{0.897}_{0.005}$ | $\mathbf{0.947}_{0.002}$ | $0.775_{0.023}$ | $0.976_{0.007}$ |
| | ABMIL | $0.889_{0.005}$ | $0.933_{0.002}$ | $\mathbf{0.731}_{0.030}$ | $0.886_{0.013}$ | $0.942_{0.003}$ | $0.752_{0.023}$ | $0.893_{0.007}$ | $0.943_{0.002}$ | $0.792_{0.022}$ | $0.956_{0.011}$ |
| | DeepGraphSurv | $0.848_{0.017}$ | $0.837_{0.020}$ | $0.673_{0.017}$ | $0.877_{0.003}$ | $0.925_{0.002}$ | $0.695_{0.007}$ | $0.870_{0.010}$ | $0.938_{0.002}$ | $0.747_{0.039}$ | $0.870_{0.070}$ |
| | CLAM | $0.674_{0.157}$ | $0.893_{0.026}$ | $0.683_{0.082}$ | $0.802_{0.054}$ | $0.930_{0.002}$ | $0.775_{0.041}$ | $0.735_{0.047}$ | $0.927_{0.001}$ | $\mathbf{0.832}_{0.030}$ | $0.960_{0.029}$ |
| | DSMIL | $0.689_{0.063}$ | $0.921_{0.008}$ | $0.672_{0.110}$ | $0.761_{0.026}$ | $0.926_{0.002}$ | $0.693_{0.036}$ | $0.792_{0.041}$ | $0.925_{0.004}$ | $0.628_{0.063}$ | $0.947_{0.085}$ |
| | PathGCN | $0.888_{0.007}$ | $0.848_{0.005}$ | $0.585_{0.180}$ | $0.890_{0.017}$ | $0.943_{0.006}$ | $0.708_{0.064}$ | $0.880_{0.023}$ | $0.945_{0.006}$ | $0.741_{0.121}$ | $0.575_{0.206}$ |
| | DFTD-MIL | $0.890_{0.045}$ | $0.940_{0.001}$ | $0.706_{0.022}$ | $0.886_{0.009}$ | $0.945_{0.002}$ | $0.720_{0.031}$ | $0.870_{0.020}$ | $0.945_{0.001}$ | $0.801_{0.015}$ | $\mathbf{0.983}_{0.010}$ |
| With global interactions | SmTAP | $0.906_{0.007}$ | $0.946_{0.003}$ | $\mathbf{0.783}_{0.056}$ | $0.893_{0.009}$ | $0.944_{0.002}$ | $\mathbf{0.805}_{0.057}$ | $0.896_{0.009}$ | $0.946_{0.004}$ | $0.754_{0.032}$ | $0.976_{0.014}$ |
| | TransMIL | $0.883_{0.008}$ | $0.933_{0.010}$ | $0.771_{0.050}$ | $0.885_{0.008}$ | $0.942_{0.002}$ | $0.791_{0.027}$ | $\mathbf{0.900}_{0.013}$ | $0.939_{0.003}$ | $0.655_{0.086}$ | $0.973_{0.018}$ |
| | SETMIL | $0.869_{0.011}$ | $\mathbf{0.974}_{0.003}$ | $0.628_{0.039}$ | $0.870_{0.008}$ | $\mathbf{0.977}_{0.005}$ | $0.657_{0.030}$ | $0.895_{0.012}$ | $\mathbf{0.970}_{0.005}$ | $0.469_{0.105}$ | $0.715_{0.155}$ |
| | GTP | $0.901_{0.008}$ | $0.949_{0.004}$ | $0.577_{0.075}$ | $\mathbf{0.896}_{0.016}$ | $0.952_{0.002}$ | $0.459_{0.056}$ | $0.890_{0.015}$ | $0.945_{0.003}$ | $0.456_{0.110}$ | $0.748_{0.118}$ |
| | IIBMIL | $0.868_{0.013}$ | $0.931_{0.004}$ | $0.641_{0.012}$ | $0.861_{0.006}$ | $0.939_{0.004}$ | $0.455_{0.042}$ | $0.897_{0.006}$ | $0.939_{0.002}$ | $\mathbf{0.791}_{0.049}$ | $0.974_{0.002}$ |
| | CAMIL | $0.889_{0.019}$ | $0.938_{0.003}$ | $0.746_{0.041}$ | $0.892_{0.010}$ | $0.941_{0.002}$ | $0.738_{0.039}$ | $0.892_{0.008}$ | $0.947_{0.004}$ | $0.772_{0.034}$ | $\mathbf{0.984}_{0.007}$ |

Table 10: Bag F1 (higher is better) for different choices of the feature extractor.

| | | ResNet18 | | | ResNet50 | | | ViT-B-32 | | ResNet50+BT |
| | | RSNA | PANDA | CAMELYON16 | RSNA | PANDA | CAMELYON16 | RSNA | PANDA | CAMELYON16 | CAMELYON16 |
|---|---|---|---|---|---|---|---|---|---|---|---|
| Without global interactions | SmAP | $0.787_{0.026}$ | $\mathbf{0.915}_{0.002}$ | $0.661_{0.056}$ | $0.788_{0.031}$ | $\mathbf{0.918}_{0.005}$ | $\mathbf{0.713}_{0.044}$ | $\mathbf{0.805}_{0.012}$ | $\mathbf{0.918}_{0.002}$ | $\mathbf{0.701}_{0.023}$ | $0.916_{0.016}$ |
| | ABMIL | $\mathbf{0.796}_{0.011}$ | $0.909_{0.001}$ | $\mathbf{0.667}_{0.022}$ | $0.800_{0.024}$ | $0.912_{0.007}$ | $0.661_{0.019}$ | $0.788_{0.023}$ | $0.912_{0.001}$ | $0.673_{0.038}$ | $0.912_{0.027}$ |
| | DeepGraphSurv | $0.719_{0.036}$ | $0.823_{0.024}$ | $0.560_{0.021}$ | $0.770_{0.013}$ | $0.905_{0.003}$ | $0.590_{0.014}$ | $0.776_{0.019}$ | $0.908_{0.004}$ | $0.606_{0.067}$ | $0.772_{0.056}$ |
| | CLAM | $0.161_{0.291}$ | $0.868_{0.034}$ | $0.485_{0.272}$ | $0.016_{0.024}$ | $0.904_{0.005}$ | $0.676_{0.041}$ | $0.000_{0.000}$ | $0.904_{0.002}$ | $0.671_{0.033}$ | $0.897_{0.012}$ |
| | DSMIL | $0.240_{0.012}$ | $0.904_{0.008}$ | $0.252_{0.239}$ | $0.374_{0.064}$ | $0.907_{0.002}$ | $0.212_{0.118}$ | $0.683_{0.036}$ | $0.902_{0.004}$ | $0.308_{0.080}$ | $0.866_{0.136}$ |
| | PathGCN | $0.782_{0.064}$ | $0.857_{0.003}$ | $0.287_{0.324}$ | $0.757_{0.089}$ | $0.915_{0.004}$ | $0.507_{0.177}$ | $0.776_{0.012}$ | $0.914_{0.006}$ | $0.606_{0.082}$ | $0.345_{0.352}$ |
| | DFTD-MIL | $0.775_{0.282}$ | $0.903_{0.002}$ | $0.576_{0.105}$ | $\mathbf{0.806}_{0.009}$ | $0.917_{0.002}$ | $0.599_{0.043}$ | $0.798_{0.024}$ | $0.914_{0.002}$ | $0.668_{0.017}$ | $\mathbf{0.937}_{0.013}$ |
| With global interactions | SmTAP | $\mathbf{0.825}_{0.026}$ | $0.917_{0.002}$ | $\mathbf{0.677}_{0.062}$ | $0.809_{0.016}$ | $0.914_{0.003}$ | $\mathbf{0.707}_{0.020}$ | $0.807_{0.032}$ | $0.914_{0.004}$ | $0.679_{0.044}$ | $\mathbf{0.948}_{0.020}$ |
| | TransMIL | $0.716_{0.031}$ | $0.895_{0.029}$ | $0.636_{0.019}$ | $0.758_{0.045}$ | $0.905_{0.013}$ | $0.635_{0.075}$ | $0.719_{0.027}$ | $0.892_{0.024}$ | $0.453_{0.098}$ | $0.911_{0.028}$ |
| | SETMIL | $0.716_{0.036}$ | $\mathbf{0.946}_{0.003}$ | $0.540_{0.024}$ | $0.734_{0.027}$ | $\mathbf{0.951}_{0.011}$ | $0.013_{0.072}$ | $0.730_{0.014}$ | $\mathbf{0.953}_{0.004}$ | $0.451_{0.085}$ | $0.471_{0.341}$ |
| | GTP | $0.805_{0.017}$ | $0.920_{0.003}$ | $0.458_{0.082}$ | $0.807_{0.019}$ | $0.923_{0.003}$ | $0.382_{0.076}$ | $0.773_{0.015}$ | $0.912_{0.003}$ | $0.384_{0.105}$ | $0.727_{0.143}$ |
| | IIBMIL | $0.621_{0.050}$ | $0.881_{0.012}$ | $0.000_{0.000}$ | $0.667_{0.011}$ | $0.889_{0.011}$ | $0.000_{0.000}$ | $0.723_{0.061}$ | $0.893_{0.008}$ | $\mathbf{0.686}_{0.016}$ | $0.922_{0.010}$ |
| | CAMIL | $0.805_{0.028}$ | $0.911_{0.004}$ | $0.649_{0.054}$ | $\mathbf{0.811}_{0.014}$ | $0.913_{0.003}$ | $0.619_{0.039}$ | $0.792_{0.013}$ | $0.917_{0.002}$ | $0.639_{0.039}$ | $0.918_{0.018}$ |

## C  Additional figures

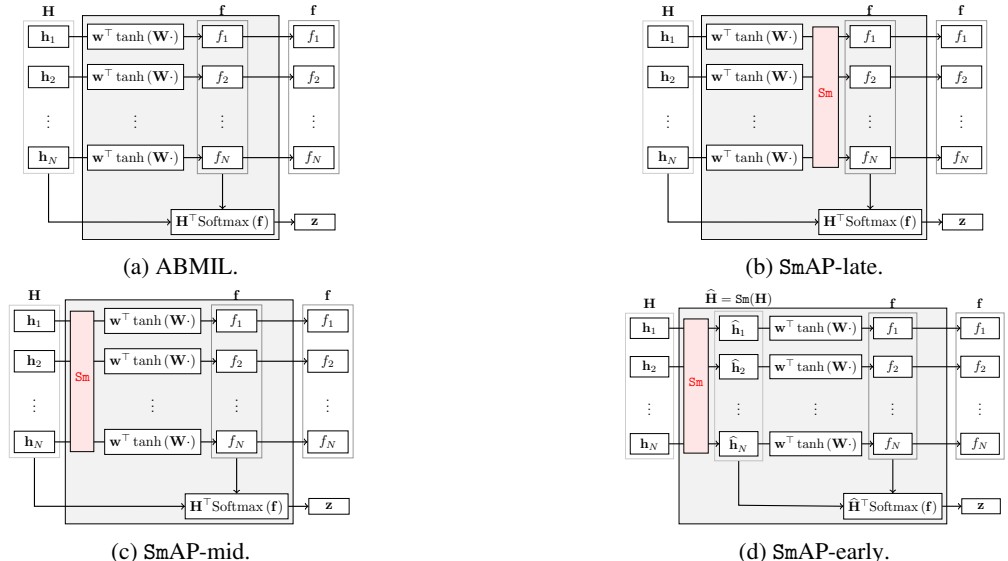

Figure 8: Graphical representation of the different variants SmAP-late, SmAP-mid, SmAP-early. The well-known ABMIL, which we build upon, is shown in (a).

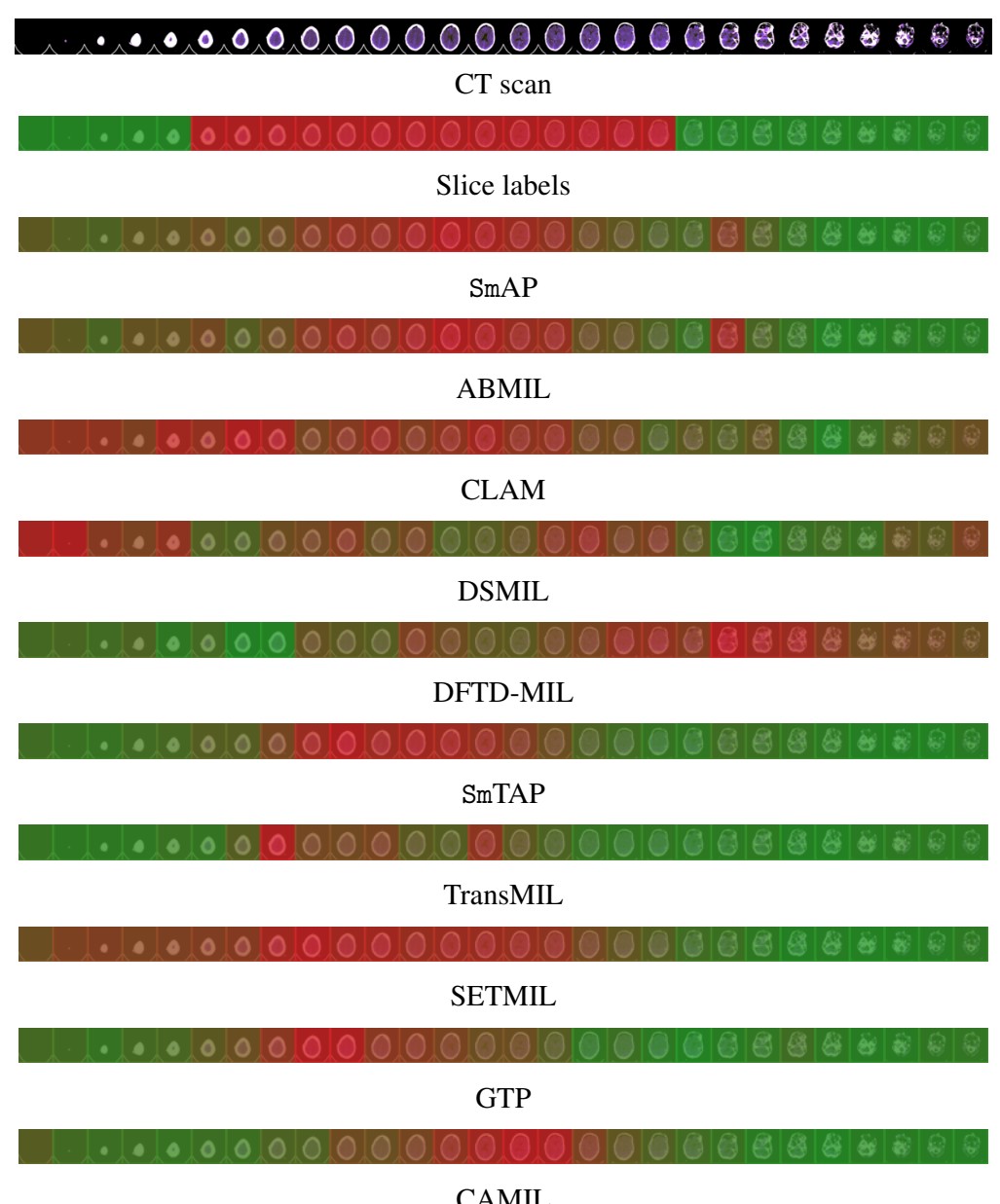

Figure 9: RSNA attention maps.

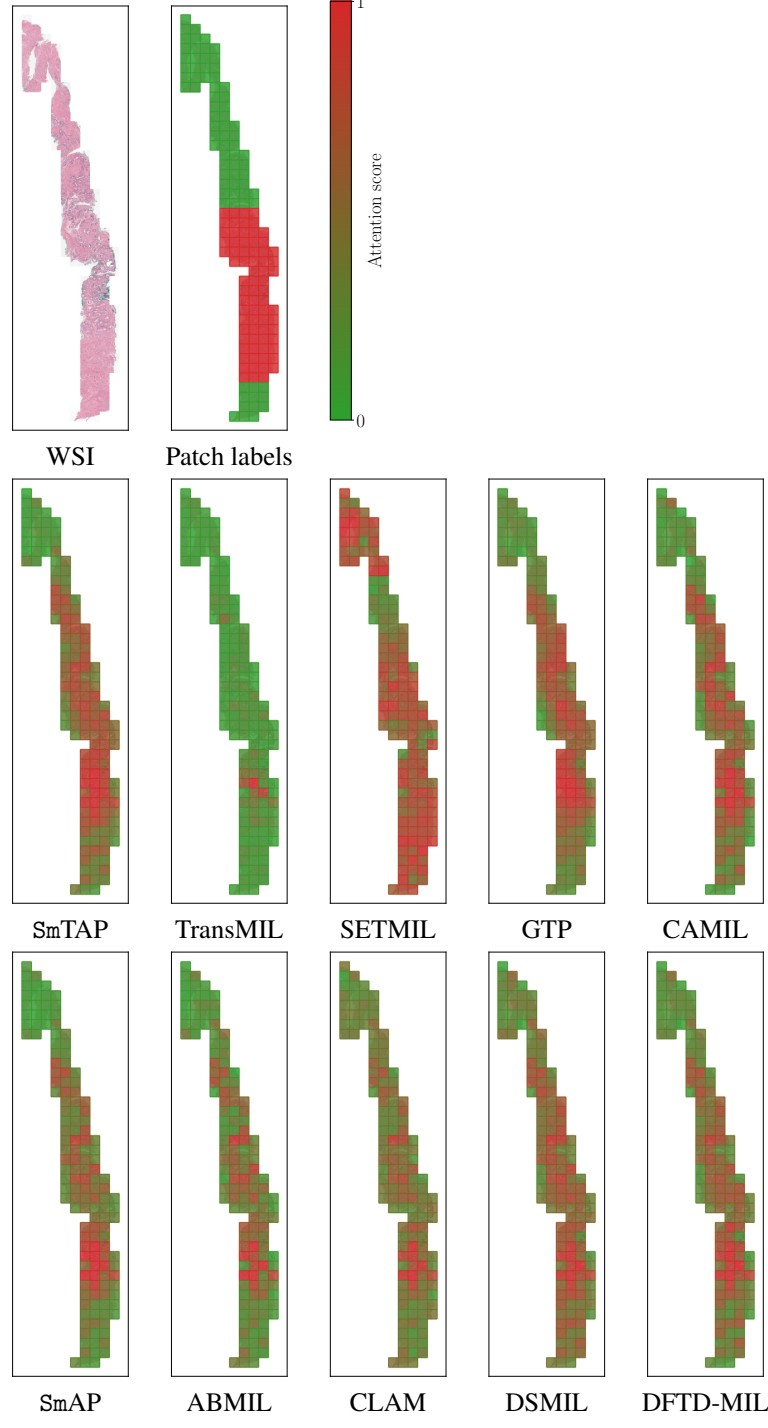

Figure 10: PANDA attention maps.

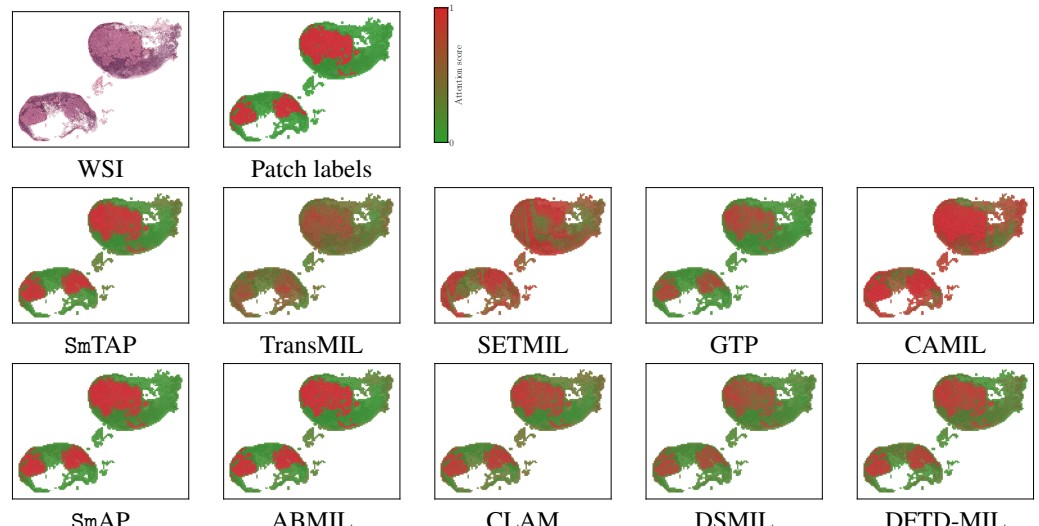

Figure 11: CAMELYON16 attention maps.

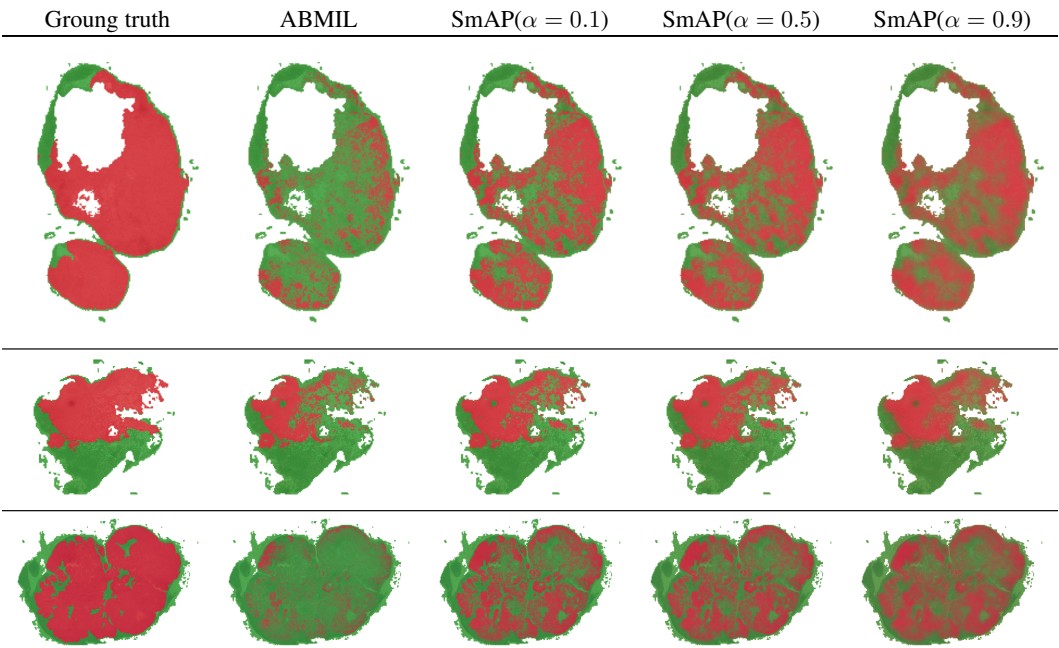

Figure 12: Ground truth and SmAP attention maps of three different WSIs from CAMELYON16. As expected theoretically, a larger $\alpha$ produces smoother attention maps.

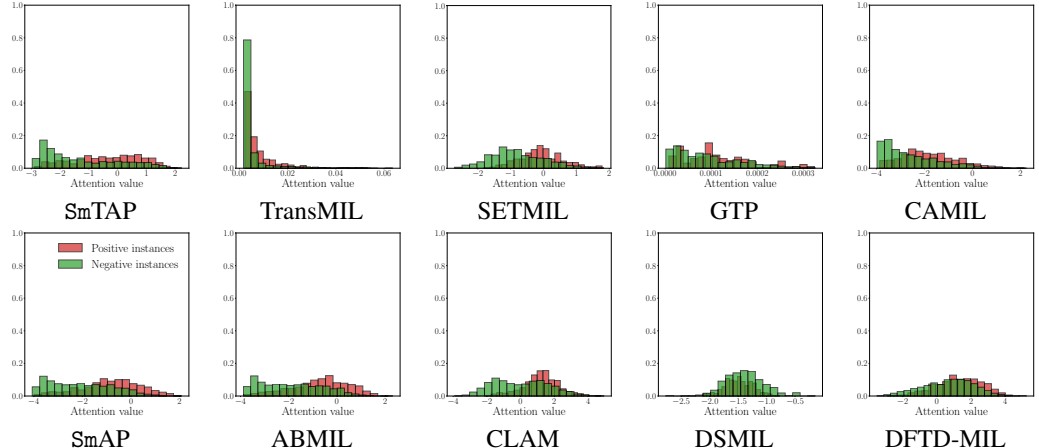

Figure 13: RSNA attention histograms.

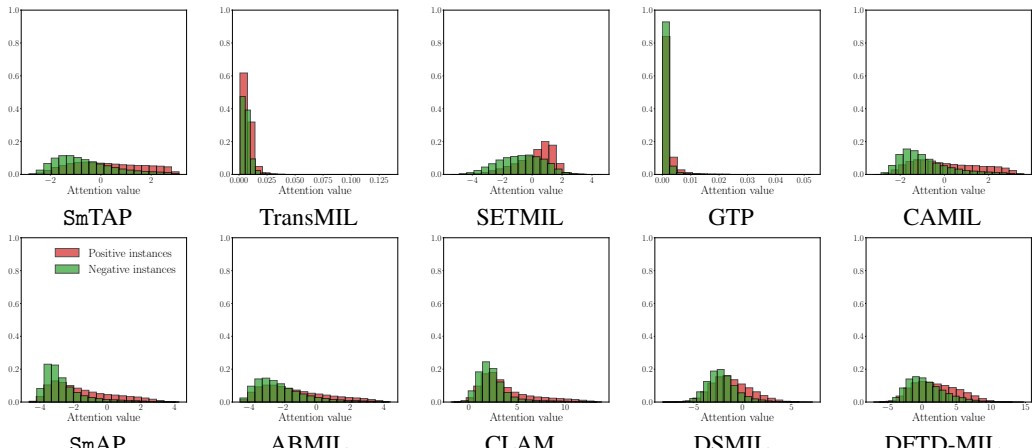

Figure 14: PANDA attention histograms.

