# OpenReview forum: "Sm: enhanced localization in Multiple Instance Learning for medical imaging classification"
_NeurIPS.cc/2024/Conference — NeurIPS 2024 poster_

### Official Review · Reviewer_uFQt · 2024-07-09

**Soundness:** 3
**Presentation:** 3
**Contribution:** 2
**Rating:** 5
**Confidence:** 4

**Summary:**

The paper is well-written and proposes a new modular mechanism that can (selectively) combine local and global interactions in a model. The authors discuss the theory behind their proposed smoothness operator and pooling mechanism and provide detailed proof for their claims. They have also evaluated their model against other methods on three different datasets (1 CT and 2 H&E) with reporting AUC and F1 score as their evaluation metrics. Their method is superior to other methods and the proper ablation study supports their claims.

**Strengths:**

The writing is concise and the contributions are clear. The author proposes a new smoothness mechanism, generally an interesting idea, to integrate local and global information. Besides the quantitative experiments, there are some qualitative experiments such as attention maps that show the model's relative importance on different regions in a slide.

**Weaknesses:**

There are three major concerns with this work:
1. The choice of encoder. The authors have used Resnet18 for RSNA and Camelyon16 and Resnet50 for PANDA. I could not find any justification for why different encoders have been used! Also, with the trend toward foundation models, transformer-based backbones are now of interest to the community. Therefore, for the sake of consistency, the author should use the same encoder for different datasets or report both resent50 and resnet18. And, for the sake of the generality of their work, they should add one encoder such as ViT or Swin as well (w/ ImageNet weights) to support the generality of their claims.

2. The body of research has been founded on local-to-global interactions. There are quite a few standard graph-based methodologies in the literature that the authors need to compare their work against. Currently, there are no representatives from those families of the MIL method in the benchmark. Two examples of such methods are (1) and (2).

3. There is a family of MIL methods in the literature that try to pseudo-label the instances during training, which is essentially equivalent to localization in this work. For instance, two of the most recent such methods are (3) and (4). The author should compare their method against these as they are essentially from the same family.

(1) Chen, Richard J., et al. "Whole slide images are 2d point clouds: Context-aware survival prediction using patch-based graph convolutional networks." Medical Image Computing and Computer Assisted Intervention–MICCAI 2021: 24th International Conference, Strasbourg, France, September 27–October 1, 2021, Proceedings, Part VIII 24. Springer International Publishing, 2021.

(2) Li, R., Yao, J., Zhu, X., Li, Y., Huang, J. (2018). Graph CNN for Survival Analysis on Whole Slide Pathological Images. In: Frangi, A., Schnabel, J., Davatzikos, C., Alberola-López, C., Fichtinger, G. (eds) Medical Image Computing and Computer Assisted Intervention – MICCAI 2018. MICCAI 2018. Lecture Notes in Computer Science(), vol 11071. Springer, Cham. https://doi.org/10.1007/978-3-030-00934-2_20

(3) Z. Shao et al., "LNPL-MIL: Learning from Noisy Pseudo Labels for Promoting Multiple Instance Learning in Whole Slide Image," 2023 IEEE/CVF International Conference on Computer Vision (ICCV), Paris, France, 2023, pp. 21438-21438, doi: 10.1109/ICCV51070.2023.01965.

(4) Ren, Q. et al. (2023). IIB-MIL: Integrated Instance-Level and Bag-Level Multiple Instances Learning with Label Disambiguation for Pathological Image Analysis. In: Greenspan, H., et al. Medical Image Computing and Computer Assisted Intervention – MICCAI 2023. MICCAI 2023. Lecture Notes in Computer Science, vol 14225. Springer, Cham. https://doi.org/10.1007/978-3-031-43987-2_54

**Questions:**

I am curious if the author expects the same performance boost obtained on top of resent, to be obtained on top of a histopathology-specific feature extractor.

**Limitations:**

It has been justified :)

---

> ### Author Rebuttal · Authors · 2024-08-06
>
> We thank Rev. uFQt for their valuable insights. Next we address their concerns.
>
> **The choice of encoder.**
> We apologize for not providing a rationale for the chosen encoder in each dataset. When we started our experiments, we fixed ResNet18 w/ Imagenet weights as the default one. This is the one used for RSNA and PANDA, since baselines obtained results comparable to those reported in previous works. However, when using this encoder in CAMELYON16, we observed that all methods were performing worse than reported in previous papers. We switched to ResNet50 with Imagenet weights, but observed a similar behavior.
> We found that some papers were using a feature extractor pre-trained with SSL on very large datasets [13, 20], so we decided to use one of these. Specifically, we used the weights provided by [17], which correspond to a ResNet50 model trained using Barlow Twins on a huge dataset of WSI patches (around 32.5 millions of patches).
> This led to the baselines obtaining comparable results to previous papers, so we decided to fix this encoder for CAMELYON16.
>
> By using two different encoders (ResNet18 and ResNet50 with Barlow Twins) we intended to show that the proposed method obtained improvements in localization/classification regardless of the type of features used. Yet, for the sake of generality, we agree that it’s interesting to report all the datasets with a wide range of encoders. Following the suggestion, we have run all the experiments using three different encoders (ResNet18, ResNet50 and ViT-B-32; all w/ ImageNet weights). Results at the instance and bag levels are included in Tables 1 and 2 of the rebuttal PDF. As a summary, the following table shows the average rank obtained by each method using the suggested encoders. The best result within each group is bolded, and the second-best is underlined. SmAP and SmTAP obtain in almost all cases the highest rank. This supports that the improvement by Sm does not depend on the used features.
>
> |||ResNet18||ResNet50||ViT-B-32||
> |-|-|-|-|-|-|-|-|
> |||Inst. rank|Bag rank|Inst. rank|Bag rank|Inst. rank|Bag rank|
> |W/o global int.|SmAP|$\mathbf{1.667}_{0.516}$|$\underline{1.917}_{0.917}$|$\mathbf{1.750}_{0.707}$|$\mathbf{1.938}_{1.084}$|$\mathbf{1.667}_{0.816}$|$\mathbf{1.500}_{1.225}$|
> ||ABMIL|$2.667_{1.366}$|$\mathbf{1.667}_{0.816}$|$3.750_{1.581}$|$3.188_{0.843}$|$4.333_{2.160}$|$3.000_{0.894}$|
> ||CLAM|$6.167_{0.983}$|$5.500_{1.225}$|$4.750_{2.053}$|$4.375_{2.134}$|$6.167_{0.983}$|$5.000_{2.449}$|
> ||DSMIL|$5.167_{0.983}$|$5.333_{1.506}$|$6.375_{0.518}$|$6.000_{0.926}$|$5.667_{1.033}$|$6.667_{0.516}$|
> ||PathGCN|$5.833_{1.472}$|$5.250_{1.605}$|$5.625_{1.302}$|$4.688_{2.017}$|$4.667_{1.633}$|$4.000_{1.549}$|
> ||DFTD-MIL|$\underline{2.000}_{0.894}$|$2.833_{1.169}$|$3.125_{1.553}$|$\underline{2.188}_{1.413}$|$\underline{2.167}_{0.983}$|$\underline{2.917}_{1.068}$|
> ||DeepGraphSurv|$4.500_{1.225}$|$5.500_{1.225}$|$\underline{2.625}_{1.188}$|$5.625_{0.916}$|$3.333_{1.211}$|$4.917_{0.376}$|
> |W/ global int.|SmTAP|$\mathbf{2.167}_{1.835}$|$\mathbf{1.667}_{1.033}$|$\mathbf{2.125}_{1.553}$|$\mathbf{1.625}_{0.518}$|$\mathbf{1.833}_{0.983}$|$\mathbf{2.500}_{0.837}$|
> ||TransMIL|$3.250_{1.255}$|$3.917_{1.201}$|$3.250_{1.035}$|$3.375_{0.916}$|$4.167_{1.329}$|$4.417_{1.908}$|
> ||SETMIL|$3.667_{0.816}$|$3.417_{1.908}$|$4.500_{0.837}$|$5.000_{0.632}$|$3.667_{1.506}$|$3.333_{1.862}$|
> ||GTP|$4.083_{1.114}$|$3.250_{1.782}$|$4.250_{1.282}$|$3.250_{2.053}$|$5.083_{1.201}$|$4.833_{1.329}$|
> ||IIBMIL|$5.167_{2.041}$|$5.667_{0.816}$|$4.250_{2.252}$|$4.875_{1.553}$|$4.167_{1.722}$|$3.250_{2.139}$|
> ||CAMIL|$\underline{2.667}_{1.751}$|$\underline{3.083}_{0.801}$|$\underline{2.250}_{1.165}$|$\underline{2.625}_{1.061}$|$\underline{2.083}_{1.281}$|$\underline{2.667}_{1.211}$|
>
> **Graph-based baselines.**
> We intended to cover these baselines by using GTP (which uses a graph convolutional network before the transformer encoder) and CAMIL (which represents the bag as a graph to mask the attention matrix). Yet, we agree that including more baselines can contextualize better. Note that the suggested methods (PathGCN and DeepGraphSurv) were proposed for survival analysis, which is a regression-related task. We have minimally adapted them for classification and run them in the three datasets, with the existing feature extractors and the new ones mentioned above. Results are included in the aforementioned tables (rebuttal PDF and summary table above).
>
> **Pseudo-labels baselines.**
> As mentioned by the reviewer, methods that assign pseudo-labels to instances can naturally address localization.
> These methods fit the general framework described in our paper (take $f_n$'s in Fig. 1a to be the pseudo-labels). We tried to cover this family in the baselines with DSMIL (which supervises pseudo-labels using the max operator and CE loss) and CLAM (which supervises them using a clustering loss). Yet, we agree it's richer to include other alternatives. Of the two methods suggested, we have included IIBMIL in the complete experimentation (see the rebuttal PDF and the summary table above). The reason for not including (3) is that it requires access to (a low percentage of) training instance labels. These labels are used to train a weak classifier to generate a set of pseudo-labels. Instance labels are unavailable in our setting during training, and the rest of methods don't use this information. Therefore, it would be a misleading/unfair comparison.
>
> **On the last question/curiosity.**
> As explained above, the results for CAMELYON16 in the paper are obtained using a histopathology-specific encoder (ResNet50 with Barlow Twins). The performance boost occurs there just as when using generic encoders.
> Our intuition is that Sm should work as long as extracted features carry valuable signal. If a poor encoder is used, then Sm may not lead to any improvement.
>
> ---
> We believe we are addressing the three raised concerns, making a more valuable work. Please let us know if something remains unclear.

---

> ### Comment · Reviewer_uFQt · 2024-08-08
> **Questions regarding the new experiments**
>
> I would like to thank the authors for extending their experiments and including new insights. I would appreciate the authors if they could provide more details regarding the following questions.
>
> It has been mentioned in line 45 that "This is a flexible module that can be used alone on top of classical MIL approaches, or in combination with transformers to also account for global dependencies." If I have understood it right, the main contribution is to have sm as a local-global module that helps to improve performance by capturing both local and global. Judging only from Table 1 and Resnet50 with ImageNet weights, how do authors justify the models without global interactions that are superior to their proposed smTAP with global interactions?
> If I am not missing anything, this is a bit contradictory to the claims of the paper.
>
> Also, comparing Resnet50+BT and Resnet50 only for the Calmelyon16 dataset, it seems that their proposed model benefits heavily from pre-trained features (improvement comes mostly from the encoder rather than the aggregation proposed). I am wondering if the authors can justify this observation. Does this come from any biological underlying meaning or can the Sm method be intuitively linked to anything?

---

> ### Comment · Area_Chair_vXTG · 2024-08-08
> **Please read the rebuttal to check if the authors addressed your concerns**
>
> Dear Reviewer uFQt,
>
> Can you have a look at the rebuttal and see if your concerns have been addressed?
>
> Best regards
> Your AC.

---

> ### Author Response · Authors · 2024-08-10
> **On the new questions**
>
> We would like to thank Reviewer uFQt for their positive comments on the new experiments and the new insights we have provided. We address the new questions below.
>
> > **Q. It has been mentioned in line 45 that "This is a flexible module that can be used alone on top of classical MIL approaches, or in combination with transformers to also account for global dependencies." If I have understood it right, the main contribution is to have sm as a local-global module that helps to improve performance by capturing both local and global. Judging only from Table 1 and Resnet50 with ImageNet weights, how do authors justify the models without global interactions that are superior to their proposed smTAP with global interactions? If I am not missing anything, this is a bit contradictory to the claims of the paper.**
>
> As we indicate in the paper, the proposed Sm operator introduces local interactions in a principled manner (through Dirichlet energy minimization with the presented theoretical guarantees). This operator can be naturally combined with self-attention layers (transformer layers) to account for both local and global interactions.
>
> As the reviewer points out, in Table 1 of the rebuttal PDF, when looking at ResNet50 with Imagenet weights, the models without global interactions (i.e., no transformer layers) do better at the instance level (localization task), with the proposed SmAP achieving the highest rank (see the table in the rebuttal to Reviewer uFQt). However, when looking at the classification task (bag level, Table 2), the models with global interactions outperform to those without global interactions, with the proposed SmTAP achieving the highest rank (see the table in the rebuttal to Reviewer uFQt).
>
> These results suggest that models without global interactions (SmAP) perform better at the instance level (localization task, Table 1), while models with global interactions perform better at the bag level (classification task, Table 2). This behaviour was also pointed out by Rev. YAE1. As we argue in the rebuttal to Reviewer YAE1, this is explained by the fact that self-attention layers of transformers obtain a richer bag representation (which leads to better bag level results). However, the signal that identifies each token in the sequence becomes weaker after each self-attention layer (the tokens become more similar). This leads to worse instance level results. Note that this phenomenon is known as over-smoothing and has been observed in (1), (2), (3) (see references below).
>
> We believe that the results in Table 1 are not contradictory with the claims in the paper, but rather reinforce them. The proposed Sm improves instance level results (localization) and remains competitive at the bag level (classification). This holds true whether it is combined with transformer layers (SmTAP) or not (SmAP).
>
> > **Q. Also, comparing Resnet50+BT and Resnet50 only for the Calmelyon16 dataset, it seems that their proposed model benefits heavily from pre-trained features (improvement comes mostly from the encoder rather than the aggregation proposed). I am wondering if the authors can justify this observation. Does this come from any biological underlying meaning or does the Sm method can be intuitively linked to anything?**
>
> As the reviewer points out, the proposed Sm benefits from SSL pre-trained features. Our intuition is that if an SSL pre-trained encoder is used, instances that are similar from a biological point of view will tend to cluster together in the feature space. In particular, instances with the same label will be clustered together, and far away from instances with a different label. Since neighboring instances are naturally (a priori) expected to have similar biological properties (e.g., the same label), the smoothness introduced by the proposed Sm will be particularly beneficial. This interesting observation is related to that in CAMIL [13], where it was shown that a SSL pre-trained encoder is crucial for the optimal performance of their neighbor-constrained attention mask.
>
> ----
>
> (1) Shi, Han, et al. "Revisiting Over-smoothing in BERT from the Perspective of Graph." International Conference on Learning Representations (ICLR). 2022.
>
> (2) Wang, Peihao, et al. "Anti-Oversmoothing in Deep Vision Transformers via the Fourier Domain Analysis: From Theory to Practice." International Conference on Learning Representations (ICLR). 2022.
>
> (3) Dong, Yihe, Jean-Baptiste Cordonnier, and Andreas Loukas. "Attention is not all you need: Pure attention loses rank doubly exponentially with depth." International Conference on Machine Learning (ICML). PMLR, 2021.

---

> ### Author Response · Authors · 2024-08-12
> **Concern whether our reply was satisfactory**
>
> In case it is necessary, we want to summarize our response to the new questions by the reviewer.
>
> > **Q. It has been mentioned in line 45 that "This is a flexible module that can be used alone on top of classical MIL approaches, or in combination with transformers to also account for global dependencies." If I have understood it right, the main contribution is to have sm as a local-global module that helps to improve performance by capturing both local and global. Judging only from Table 1 and Resnet50 with ImageNet weights, how do authors justify the models without global interactions that are superior to their proposed smTAP with global interactions? If I am not missing anything, this is a bit contradictory to the claims of the paper.**
>
> Instance level results worsen with transformers. This is not a consequence of the proposed Sm, but of the oversmoothing phenomenon observed in transformers. This has been studied in the works referenced in our previous response.
>
> The proposed method without a transformer outperforms methods that do not a transformer; and the proposed method with a transformer outperforms methods that use a transformer.
>
>
> > **Q. Also, comparing Resnet50+BT and Resnet50 only for the Calmelyon16 dataset, it seems that their proposed model benefits heavily from pre-trained features (improvement comes mostly from the encoder rather than the aggregation proposed). I am wondering if the authors can justify this observation. Does this come from any biological underlying meaning or does the Sm method can be intuitively linked to anything?**
>
> Note that every method has access to the same features, and every method benefits from using them (see bag level results in Table 2). That our method benefits at the instance level from using, we interpret it as a sign that local dependencies indeed play an important role in MIL localization.
>
> Please let us know if you have any questions so that we can respond in a timely manner.
>
> We look forward to your response,
>
> Authors

---

> > ### Comment · Reviewer_uFQt · 2024-08-12
> >
> > I appreciate the authors' effort in providing further details and explanations for my questions. The response provided addressed my concern partly, yet I agree with Reviewer YAE1 regarding the improvement being marginal. However, given the efforts in providing newer benchmarks and also considering that the content needs to be updated with consistent backbones for different datasets (as provided in the previous reply). Therefore, I would raise my rating to Borderline Accept).

---

### Official Review · Reviewer_YAE1 · 2024-07-12

**Soundness:** 3
**Presentation:** 2
**Contribution:** 3
**Rating:** 5
**Confidence:** 4

**Summary:**

The paper proposed a multi-instance learning approach. The basic idea is to make use of the spatial dependency between training samples. A smoothing operator was proposed to regularize the attention matrix with respect to inter-sample similarity (to my understanding), which the authors claimed to improved both quality of localization and classification tasks.

**Strengths:**

The idea seems unique and based, but more justifications are merited.

**Weaknesses:**

1. Although the smoothness property do exist in multi-instance learning as the authors claimed and may introduce useful information. It is still unclear why imposing it in the neural network helps with classification. It doesn't seem to serve as an induction bias from its looks.
2. Actually only 3 dataset are tested (though each with different variants), and the results are not so persuasive. Especially for the classification tasks that the authors want to credit, they failed to get 1st place in 2/3 of the AUC or F1 scores.

**Questions:**

1. The overall principle of the method need rephasing, i.e., how introduction the SM transformation (instead of penalty) improves classification?
2. The smoothness assumption seems to be a property connected with localization. However, in 4.1-4.3 and Fig.3 the authors mainly present the SM operator for  classification. Yet in the experiment part, it seem that adding SM had more advantages in localization (but not sure how is it achieved?). The authors need to clarify the relationship between the two tasks.
3. In Tab. 1, why the performance of the 'with global interactions' variants worse than the 'without' one in localization, but it is opposite in the classification (Tab. 2)?

**Limitations:**

The method required strict ordering (or spatial information) or 3d-to-2d samples, which is quite a strong demand but it is not discussed in the limitations.

---

> ### Author Rebuttal · Authors · 2024-08-05
>
> Thanks for your feedback. First we provide a general clarification. Then we address the rest of points.
>
> ---
> **GENERAL CLARIFICATION**
>
> It is unfortunate that the concepts of classification and localization used in the manuscript are not found clear. These concepts are taken from the MIL literature, especially from the medical-related one. Next, we try to clarify them.
>
> Given a test medical image (e.g., a WSI image), our goal is twofold: 1) make a global prediction of whether the WSI contains tumor or not, and 2) predict which patches are the tumorous ones. The former task is called **classification**, and the latter **localization**. Notice that, from the ML viewpoint, both are classification tasks.
>
> **Labels available during training**. The training dataset only contains labels at the WSI level. We know whether each training WSI is cancerous or not, but not which patches contain the tumor. This is exactly the MIL paradigm: WSIs are bags and their patches are instances. Under this MIL setting (bags $\equiv$ WSI, instances $\equiv$ patches), the classification/localization results are usually called bag-/instance-level results, respectively.
>
> **Making predictions at instance-level (localization) although only bag-level labels are available during training**.
> This is precisely what MIL methods address. Generally, the idea is to process the instances inside a bag and then aggregate them through some instance-wise attention values ($f_n$ in Figure 1), which reflect the relevance of each instance. Then, these values are used for instance-level prediction.
>
> **We focus on localization task**.
> Current approaches are good at the classification task, but their localization results are comparatively limited. Thus, our goal is to enhance localization while staying competitive in classification. This focus is reflected in the paper title “enhanced localization…”. Since both tasks are related, we find that our approach is also slightly superior in classification. We hope this clarifies this sentence by the reviewer: “Especially for the classification tasks that the authors want to credit”.
>
> **The rationale behind our proposal to enhance localization**.
> Current MIL methods are designed to target the classification task. In terms of localization, they pinpoint some regions of interest, but a systematic instance-level evaluation reveals low performance metrics. We find these methods do not account for spatial correlations among instances (patches) or, if they do (as CAMIL or GTP), they consider them only to obtain the bag-level aggregation, but not for instance-level predictions. Thus, our idea is to encode these local dependencies into the MIL model in a principled way (through Dirichlet energy minimization with theoretical guarantees). Intuitively, these local correlations tend to uniformize instance-level predictions, avoiding isolated false positives or false negatives.
>
> ---
> **REMAINING POINTS**
>
> **Why imposing smoothness helps**.
> The basic idea is that in real-world medical images the instance labels tend to have a smooth distribution. It is not realistic to have isolated negative patches inside positive regions and the other way around. This uniformity is the inductive bias introduced by our operator. Since we are acting at instance-level, this has a clear effect in localization, see Table 1 in the paper. As a byproduct, since the WSI prediction is aggregated from instance-level info, it also improves the classification task (Table 2), although the margin is lower.
>
> **Persuasiveness of results and number of datasets.**
> Recall we expect the main improvement to occur at localization (Table 1). Regarding classification results (Table 2) the differences are lower as expected, but the proposed methods are still the best-performing ones.
> Regarding the number of datasets, the comprehensive evaluation of localization requires that the test set is completely labeled (e.g., every patch in the WSIs). This restricts the number of datasets available. Most recent MIL papers in top venues use three or fewer datasets. The papers [13], [37], [38], [29] (ICLR 24, MICCAI 22, IEEE TMI 22, NeurIPS 21) contain, respectively, 2, 2, 2 and 3 datasets.
>
> **Penalty as an alternative to the proposed operator.**
> Introducing a penalty term in the loss to favor smoothness is a natural alternative to the proposed operator. This is also commented by Rev. jUey. Please see the response to them (paragraph headed "Loss-based strategy for smoothing"), which includes an ablation.
>
> **W/ and w/o global interactions behave differently in Tables 1-2**. Yes, this is an interesting observation that points out a limitation of transformers in MIL. Transformers excel at aggregating/encoding a complex sequence (e.g. a WSI) to make a global prediction, which is the goal of the classification task. That’s why methods with global interactions (mainly transformers) outstand in Table 2. However, they yield lower performance at the localization task (instance-level predictions, Table 1). This is because the self-attention layers of transformers lead to a loss of signal in the individual tokens of the sequence (e.g., the WSI patches). After such layers we obtain a valuable aggregation, but instance-level information has been degraded. Indeed, the proliferation of transformers in MIL is yet another sign that the literature has so far overlooked the localization task, which is key for real-world deployment.
>
> **Limitations on the spatial information**. We acknowledge that this method is not applicable to MIL problems where there is not a spatial structure of the bag. However, medical images (e.g. WSIs and CT scans) inherently possess this structure. To handle it efficiently, we use the torch.sparse API (a common approach in GNNs). This way, the memory overhead is minimal compared to the bag size.
>
> ---
>
> We hope the reviewer reconsiders the evaluation of our contribution in the light of this clarification. Please let us know if something remains unclear.

---

> ### Comment · Area_Chair_vXTG · 2024-08-08
> **Please read the rebuttal to check if the authors addressed your concerns**
>
> Dear Reviewer YAE1,
>
> Can you have a look at the rebuttal and see if your concerns have been addressed?
>
> Best regards
> Your AC.

---

> > ### Comment · Reviewer_YAE1 · 2024-08-09
> > **reviewer feedback**
> >
> > Thanks for the authors' effort to present additional results and make clarifications.   I raised the score despite the concern that this still seems to be a marginal improvement.

---

### Official Review · Reviewer_jUey · 2024-07-13

**Soundness:** 3
**Presentation:** 3
**Contribution:** 3
**Rating:** 6
**Confidence:** 4

**Summary:**

The authors propose a technique to improve the localization capabilities of the current MIL, especially for CAD models that perform CT and WSI analysis. The method is based on seeing the attention attributed to each patch as a graph and minimizing its Dirichlet energy, promoting smooth transitions on the attention values of neighbor patches, which is in accordance with the assumption that neighboring patches share the same labels.

**Strengths:**

The work is well-developed and well written, with a methodology explained through concise equations and theorems that guide the reader through the development of the idea. The results validate what is proposed, but minor adjustments and additional information will make the work even more valuable.

**Weaknesses:**

The method introduces the smooth operator on top of the existing ABMIL model. The proposed approach improves the performance of ABMIL on the CAMELYON dataset, both on localization and classification. Nevertheless, when analyzing the attention maps provided in the appendix, the differences between ABMIL and smAP are minimal. Although we have a numerical evaluation of the Dirichlet energy, we cannot tell if the drop in energy is significant only by these numbers, so this visual empiric evaluation is important for a better insight regarding the developed work.

Observing the appendix figures, ABMIL already performs well in matching its attention to the tiles annotations. I suggest the authors apply the smooth operator on top of other models that did not perform well on CAMELYON for this task, such as CLAM, CAMIL, and SETMIL, to evaluate how minimizing the Dirichlet energy of the attention improves the localization task quantitatively and qualitatively.

I suggest the authors perform an ablation study to evaluate how minimizing the Dirichlet energy compares to other smoothing strategies. For instance, summing the variance of the attentions to the model’s loss may provide similar results as minimizing the Dirichlet energy.

The analysis of the alpha parameter is great, but again, a qualitative approach is valuable for this case. Seeing how the attention map changes as the parameter alpha increases is highly valuable for the reader.

Lines 127 to 131: trying to debate equations not present in the paper is not good for the reader. I suggest writing the debated equations in the article or explaining the arguments without mentioning them.

**Questions:**

The formulation seems to assume bags of equal size, which is not the case in MIL (in general).

The proposed method (loss) may have the drawback of over-smoothing the transition between tiles, making the localization regions blurry. Is this true? Could this be overcome?

**Limitations:**

yes

---

> ### Author Rebuttal · Authors · 2024-08-01
>
> We thank Rev. jUey for their positive and valuable feedback. Next we address their concerns.
>
> **Visual comparison between ABMIL and SmAP.**
> As pointed out by Rev. jUey, the fact that the Dirichlet energy decreases when using the proposed operator (Table 3 in the paper) does not imply that the instance-level performance (localization task) or the bag-level one (classification task) are enhanced.
> Such enhancement can only be assessed by computing performance metrics both at instance and bag level, which is done in Tables 1 and 2 in the paper, respectively. Those tables show that SmAP outperforms ABMIL, especially at localization (see e.g. CAMELYON16, where they obtain 0.96 vs 0.82 in AUROC and 0.84 vs 0.77 in F1-score). Although the attention maps shown for SmAP and ABMIL in CAMELYON16 are very similar, please note that this is the visualization of *just one WSI*, which was selected at random (no cherry-picking). Instead, numerical results in Tables 1-2 in the paper consider all the 130 WSIs in the test set, not only one. Yet, we agree it can be misleading to use only one test WSI for visualization. Thus, in the rebuttal PDF we include three different test WSIs of CAMELYON16, where the overall enhancement of SmAP over ABMIL can be appreciated. It is interesting to see how SmAP better defines the positive (red) region thanks to the smooth operator.
>
> **Applying Sm on top of other baselines.**
> As explained above, if we look at aggregated results of Table 1 in the paper instead of a particular WSI, ABMIL is not the best-performing among the baselines. Yet, it is worth exploring how other approaches behave when being combined with Sm. We have done it for the rest of the baselines w/o global interactions, i.e. CLAM, DSMIL, and DFTD-MIL, using CAMELYON16. Notice that adding Sm on top of the baselines w/ global interactions is not as natural, as those methods carry their own mechanisms to account for interactions. The results are in the table below. The improvements when adding Sm are in bold.
>
> |Model|Sm|Instance||Bag||
> |-|-|-|-|-|-|
> |||AUROC|F1-score|AUROC|F1-score|
> |CLAM|No|$0.849_{0.044}$|$0.821_{0.046}$|$0.960_{0.029}$|$0.897_{0.012}$|
> ||Yes|$\mathbf{0.928}_{0.028}$|$\mathbf{0.873}_{0.018}$|$\mathbf{0.966}_{0.007}$|$0.889_{0.017}$|
> |DSMIL|No|$0.76_{0.078}$|$0.654_{0.203}$|$0.947_{0.085}$|$0.866_{0.136}$|
> ||Yes|$\mathbf{0.960}_{0.013}$|$\mathbf{0.776}_{0.088}$|$\mathbf{0.967}_{0.011}$|$\mathbf{0.919}_{0.018}$|
> |DFTD-MIL|No|$0.884_{0.002}$|$0.742_{0.040}$|$0.983_{0.010}$|$0.937_{0.013}$|
> | |Yes|$0.884_{0.183}$|$\mathbf{0.836}_{0.222}$|$0.978_{0.158}$|$0.903_{0.183}$|
>
> The conclusion is similar to the main manuscript: instance-level performance is enhanced (greatly in some cases, e.g. an increase from 0.76 to 0.96 in AUROC for DSMIL), whereas bag-level results are competitive. The decrease in bag-level for DFTD-MIL is explained because this method randomly splits each bag into different *chunks*, which may lead to the loss of local interactions exploited by Sm (e.g., if two adjacent instances end in different chunks).
>
> **Loss-based strategy for smoothing.**
> Introducing a penalty term in the loss to favor smoothness is a natural alternative to the proposed operator. However, there is a key difference, since the use of a penalty term does not modify the model architecture. The penalty term favors that the learned weights encode such property, but it is not encoded explicitly in the model. For instance, at test stage, the penalty term is not used. Still, we find interesting to include this ablation study. The results are in Table 1 in the global response above, as this was also asked by Rev. YAE1. We conclude that, although differences are not large, Sm obtains superior performance.
>
> **Qualitative assessment of $\alpha$.**
> In the rebuttal PDF, we include the attention map by SmAP with three values of $\alpha$ for the same WSIs used before. As theoretically expected, higher $\alpha$ introduces more smoothness. Yet, as reported in the paper, the performance difference among $\alpha$'s is small, and the largest difference is against ABMIL (which treats every instance independently).
>
> **On lines 127-131.**
> Thanks for the suggestion, which helps to clarify the paper. In the revised version we have included the equations explicitly, as this is an important insight into why previous approaches that account for local interactions obtain low performance at instance level.
>
> **Bags seem to be of equal size.**
> Thanks for pointing this out, the formulation in lines 94-95 can be misleading. No, bags do not need to be equal in size. Indeed, in the description of the RSNA dataset (Appendix B.1), we mention that there are 492 scans (bags) whose size varies from 24 to 57. The same applies to the rest of the datasets, where different WSIs have different number of patches. We have clarified this explicitly in lines 94-95.
>
> **Over-smoothing.**
> Thank you, this is a common problem when introducing spatial correlations, not only in MIL but also in very related image processing techniques (e.g. deblurring). To address it, in our paper we use a “soft” adjacency matrix, following previous approaches, e.g. [13]:  if two instances $(i,j)$ are not adjacent, then $A_{ij}=0$. If they are adjacent, the value is $A_{ij} = \exp\left(-P^{-1}\sqrt{|| \mathbf{h}\_i  -\mathbf{h}\_j ||}\right),$
> where $\mathbf{h}\_k \in \mathbb{R}^P$ is the embedding of each patch through the feature extractor.
> This $A_{ij}$ is in the interval $[0,1]$, and increases as the embeddings become closer.
> Note that the degree of smoothness depends on the *visual similarity of the instances*, avoiding unintended over-smoothing when positive and negative instances are not close.
> We have noticed that this was not specified in the paper (it could only be seen in the code). We have included it in the revised version.
>
> ---
> We hope these responses and additional materials make the work more valuable to the reviewer.

---

> ### Comment · Area_Chair_vXTG · 2024-08-08
> **Please read the rebuttal to check if the authors addressed your concerns**
>
> Dear Reviewer jUey,
>
> Can you have a look at the rebuttal and see if your concerns have been addressed?
>
> Best regards
> Your AC.

---

> ### Author Response · Authors · 2024-08-12
> **Have your concerns been addressed?**
>
> Dear Reviewer jUey,
>
> We are wondering if you need any additional clarification. Please let us know so that we can respond to you in a timely manner.
>
> We look forward to your response,
>
> Authors

---

### Author Rebuttal · Authors · 2024-08-06

## **GLOBAL RESPONSE**

We **thank the three reviewers** for taking their time to read our work, providing constructive and valuable feedback. We appreciate it.
We are happy that **the feedback is in overall positive**: "well-developed and well written", "minor adjustments and additional information will make the work even more valuable", "the idea seems unique and based", "the writing is concise and the contributions are clear", "generally an interesting idea" etc.

**Several concerns** have been raised too. **Most of them are specific points** which we believe we have been able to address and/or clarify in the responses. We hope the reviewers find it valuable and they consider raising their scores. Please do not hesitate to request any additional clarification during the discussion phase.

The response is organized as follows:

* This **global response** contains a general overview as well as a table requested by more than one reviewer. Each reviewer will be pointed to these tables from their individual response.
* We also attach a **one-page rebuttal PDF file** to this global response. It contains only figures and tables. Each reviewer will be pointed to these items from their individual response.
* We provide an **individual response** for each reviewer.

After reading this general response, we encourage each reviewer to read their individual one, which will include or point to supporting tables and figures.

---

### **GENERAL OVERVIEW**

The following list provides a brief summary of our response to the main concerns expressed by the reviewers. We have
* Added three new baselines. [Reviewer uFQt]
* Extended all the experimentation using the three same encoders for all the datasets. [Reviewer uFQt]
* Provided a thorough clarification for the proposed approach as well as for key concepts in medical-related MIL (classification and localization tasks). [Reviewer YAE1]
* Included an ablation against an alternative way of favoring smoothness. [Reviewers YAE1 and jUey]
* Included additional attention maps to provide a more representative visualization. [Reviewer jUey]
* Evaluated how the proposed operator performs when combined with other baselines. [Reviewer jUey]

---

TABLE 1: Comparing the proposed smoothing operator with a penalty-based mechanism to favor smoothness. Requested by Revs. jUey and YAE1.

|||RSNA||PANDA||CAMELYON16||
|-|-|-|-|-|-|-|-|
|||Inst. AUROC $(\uparrow)$|Bag AUROC $(\uparrow)$|Inst. AUROC $(\uparrow)$|Bag AUROC $(\uparrow)$|Inst. AUROC $(\uparrow)$|Bag AUROC $(\uparrow)$|
|W/o global int.|SmAP|$\mathbf{0.798}_{0.033}$|$0.888_{0.005}$|$\mathbf{0.799}_{0.005}$|$\mathbf{0.943}_{0.001}$|$0.961_{0.007}$|$\mathbf{0.965}_{0.007}$|
||ABMIL+PENALTY|$0.782_{0.050}$|$\mathbf{0.889}_{0.043}$|$0.780_{0.003}$|$0.935_{0.001}$|$\mathbf{0.979}_{0.013}$|$0.963_{0.012}$|
|W/ global int.|SmTAP|$\mathbf{0.767}_{0.046}$|$\mathbf{0.906}_{0.007}$|$\mathbf{0.790}_{0.007}$|$0.946_{0.003}$|$\mathbf{0.789}_{0.008}$|$0.976_{0.014}$|
||TAP+PENALTY|$0.737_{0.045}$|$0.905_{0.005}$|$0.772_{0.011}$|$\textbf{0.947}_{0.001}$|$0.769_{0.099}$|$\mathbf{0.988}_{0.004}$|

---

### Decision · Program_Chairs · 2024-09-25

**Decision:**

Accept (poster)

**Comment:**

This paper introduces an approach to improve the localization task of multiple instance learning (MIL) methodologies, particularly the ones applied to CT and WSI. Relying on a spatial dependency between training samples, the paper presents a smoothing operator to regularize the attention matrix with respect to inter-sample similarity. The paper also presents a theoretical analysis of the method. The proposed method shows better results than competing approaches and the ablation study supports its claims. The paper received an average score of 5.33 (5,5,6), and the main strengths found are:  1) paper is well-written and concise, 2) claims are well supported by experimental results, 3) idea explored in the paper is interesting, and 4) qualitative experiments show the model’s advantages.  The reviewers also identified the following weaknesses: 1) negligible differences between ABMIL and the proposed method; 2) missing experiment on the application of the proposed smooth operator on top of other methods (e.g., CLAM, CAMIL, SETMIL); 3) missing ablation study on a comparison between the minimisation of Dirichlet energy and other smoothing strategies; 4) missing qualitative experiment on the role of alpha parameter; 5) unclear role of the smoothness property for the classification from a neural network; 6) unconvincing results using only 3 datasets, where the proposed method failed to get first place in 2/3 of the AUC or F1 scores; 7) unclear justification for different encoders for different datasets; 8) missing comparisons with graph-based methodologies; and 9) missing comparison with MIL methods that rely on pseudo labels. There was a fair bit of discussion during the rebuttal, and the only remaining issue of the paper is that the performance improvement compared to competing approaches is marginal. I support the publication of this paper, but encourage the authors to address the issues raised during this rebuttal phase.